# Convex-Concave 0-Sum Markov Stackelberg Games

**Denizalp Goktas**
Brown University, Computer Science
`denizalp_goktas@brown.edu`

**Arjun Prakash**
Brown University, Computer Science
`arjun_prakash@brown.edu`

**Amy Greenwald**
Brown University, Computer Science
`amy_greenwald@brown.edu`

## Abstract

Zero-sum Markov Stackelberg games can be used to model myriad problems, in domains ranging from economics to human robot interaction. We develop a policy gradient method which we prove solves these games in continuous state, continuous action settings, using noisy gradient estimates computed from observed trajectories of play. When the games are convex-concave, we prove that our algorithm converges to Stackelberg equilibrium in polynomial time. We also prove that reach-avoid problems are naturally modeled as convex-concave zero-sum Markov Stackelberg games, and show experimentally that Stackelberg equilibrium policies are more effective than their Nash counterparts in these problems.[1]

## 1 Introduction

Markov games [28, 65, 70] are a generalization of Markov decision processes (MDPs) comprising multiple players simultaneously making decisions over time, collecting rewards along the way depending on their collective actions. They have been used by practitioners to model many real-world multiagent planning and learning environments, such as autonomous driving [31, 59], cloud computing [77], and telecomunications [3]. Moreover, theoreticians are beginning to formally analyze policy gradient methods, proving polynomial-time convergence to optimal policies in MDPs [2, 16], and to Nash equilibrium policies [53] in zero-sum Markov games [24], the canonical solution concept. While Markov games are a fruitful way to model some problems (e.g., robotic soccer [46]), others, such as reach-avoid [48], may be more productively modeled as sequential-move games, where some players commit to moves that are observed by others, before they make their own moves. To this end, we study two-player zero-sum Markov Stackelberg [74] (i.e., sequential-move) games. While polynomial-time value-iteration (i.e., planning) algorithms are known for these games assuming discrete states [36], we develop a policy gradient method that converges to Stackelberg equilibrium in polynomial time in continuous state, continuous action games, using noisy gradients based only on observed trajectories of play. Furthermore, we demonstrate experimentally that Stackelberg equilibrium policies are more effective than their Nash counterparts in reach-avoid problems.

A *(discounted discrete-time) zero-sum Markov Stackelberg game* [36] is played over an infinite horizon $t = 0, 1, \ldots$ between two players, a leader and a follower. The game starts at time $t = 0$, at some initial state $S^{(0)} \sim \mu \in \Delta(\mathcal{S})$ drawn randomly from a set of states $\mathcal{S}$. At each time step $t = 1, 2, \ldots$, the players encounter a state $\boldsymbol{s}^{(t)} \in \mathcal{S}$, where the leader takes its action $\boldsymbol{a}^{(t)}$ first, from its action space $\mathcal{A}(\boldsymbol{s}^{(t)})$, after which the follower, having observed the leader's action, makes it own move $\boldsymbol{b}^{(t)}$, chosen from a feasible subset $\mathcal{C}(\boldsymbol{s}^{(t)}, \boldsymbol{a}^{(t)})$ determined by the leader's action $\boldsymbol{a}^{(t)}$

---

[1]A full and current version of the paper can be found at: `https://arxiv.org/abs/2401.12437`

of its action space $\mathcal{B}(\boldsymbol{s}^{(t)})$.[2] After both players have taken their actions, they receive respective rewards, $-r(\boldsymbol{s}^{(t)}, \boldsymbol{a}^{(t)}, \boldsymbol{b}^{(t)})$ and $r(\boldsymbol{s}^{(t)}, \boldsymbol{a}^{(t)}, \boldsymbol{b}^{(t)})$. The game then moves to time step $t + 1$ and transitions either to a new state $S^{(t+1)} \sim p(\cdot \mid \boldsymbol{s}^{(t)}, \boldsymbol{a}^{(t)}, \boldsymbol{b}^{(t)})$ with probability $\gamma$, called the *discount factor*, or the game ends with the remaining probability. Each player's goal is to play a (potentially history-dependent) *policy* that maximizes its respective *expected (cumulative discounted) payoffs*, $-\mathbb{E}\left[\sum_{t=0}^{\infty} \gamma^t r(S^{(t)}, A^{(t)}, B^{(t)})\right]$ and $\mathbb{E}\left[\sum_{t=0}^{\infty} \gamma^t r(S^{(t)}, A^{(t)}, B^{(t)})\right]$.[3]

In zero-sum Markov Stackelberg games, when the reward function $(\boldsymbol{a}, \boldsymbol{b}) \mapsto r(\boldsymbol{s}, \boldsymbol{a}, \boldsymbol{b})$ is continuous and bounded, for all $\boldsymbol{s} \in \mathcal{S}$, and the correspondence $\boldsymbol{a} \rightrightarrows \mathcal{C}(\boldsymbol{s}, \boldsymbol{a})$ is continuous, as well as non-empty- and compact-valued, a *recursive* (or *Markov perfect*) [49] *Stackelberg equilibrium* is guaranteed to exist [36], meaning a *stationary policy profile* (i.e., a pair of mappings from states to the actions of the leader and the follower, respectively) specifying the actions taken at each state s.t. the leader's policy maximizes its expected payoff assuming the follower best responds, while the follower indeed best responds to the leader's policy. In other words, the aforementioned assumptions guarantee the existence of a *policy profile* $\boldsymbol{\pi}^* \doteq (\boldsymbol{\pi}_{\boldsymbol{a}}^*, \boldsymbol{\pi}_{\boldsymbol{b}}^*)$, with $\boldsymbol{\pi}_{\boldsymbol{a}}^* : \mathcal{S} \to \mathcal{A}$ and $\boldsymbol{\pi}_{\boldsymbol{b}}^* : \mathcal{S} \to \mathcal{B}$, that solves the following *coupled* min-max optimization problem:

$$\min_{\substack{\boldsymbol{\pi}_{\boldsymbol{a}} : \mathcal{S} \to \mathcal{A}}} \max_{\substack{\boldsymbol{\pi}_{\boldsymbol{b}} : \mathcal{S} \to \mathcal{B}: \\ \forall \boldsymbol{s} \in \mathcal{S}, \boldsymbol{\pi}_{\boldsymbol{b}}(\boldsymbol{s}) \in \mathcal{C}(\boldsymbol{s}, \boldsymbol{\pi}_{\boldsymbol{a}}(\boldsymbol{s}))}} \mathbb{E}\left[\sum_{t=0}^{\infty} \gamma^t r(S^{(t)}, \boldsymbol{\pi}_{\boldsymbol{a}}(S^{(t)}), \boldsymbol{\pi}_{\boldsymbol{b}}(S^{(t)}))\right] , \qquad (1)$$

where the expectation is with respect to $S^{(0)} \sim \mu$ and $S^{(t+1)} \sim p(\cdot \mid \boldsymbol{s}^{(t)}, \boldsymbol{\pi}_{\boldsymbol{a}}(S^{(t)}), \boldsymbol{\pi}_{\boldsymbol{b}}(S^{(t)}))$. The problem is "coupled" because the players' actions sets constrain one another; in particular, the set of actions available to the follower at each state is determined by the leader's choice.

In spite of multiple compelling applications, including autonomous driving [29, 43], reach-avoid problems in human-robot interaction [10], robust optimization in stochastic environments [15], and resource allocation over time [36], very little is known about computing recursive Stackelberg equilibria in zero-sum Markov Stackelberg games. A version of value iteration is known to converge in polynomial time when the state space is discrete [36], but this (planning) method becomes intractable in large or continuous state spaces. Furthermore, nothing is known, to our knowledge, about *learning* Stackelberg equilibria from observed trajectories of play. We develop an efficient policy gradient method for convex-concave zero-sum Markov Stackelberg games, and we show that reach-avoid problems naturally lie in this class of games.

**Contributions.** Equation (1) reveals that the problem of computing Stackelberg equilibria in zero-sum Markov Stackelberg games is an instance of a coupled min-max optimization problem. Goktas and Greenwald [33] studied coupled min-max optimization problems assuming an *exact* first-order oracle, meaning one that returns a function's exact value and gradient at any point in its domain. As access to an exact oracle is an unrealistic assumption in Markov games, we develop a first-order method for solving these problems, assuming access to a *stochastic* first-order oracle, which returns noisy estimates of a function's value and gradient at any point in its domain. We show that our method converges in polynomial-time (Theorem 3.1) in a large class of coupled min-max optimization problems, namely those which are convex-concave.

We then proceed to study zero-sum Markov Stackelberg games, providing sufficient conditions on the action correspondence $\mathcal{C} : \mathcal{S} \times \mathcal{A} \to \mathcal{B}$, the rewards $r : \mathcal{S} \times \mathcal{A} \times \mathcal{B} \to \mathbb{R}$, and the transition probabilities $p : \mathcal{S} \times \mathcal{S} \times \mathcal{A} \times \mathcal{B} \to \mathbb{R}_+$ to guarantee that the game is convex-concave. Furthermore, we develop a policy gradient algorithm that provably converges to Stackelberg equilibrium in polynomial time when such games are convex-concave (Theorem 4.1), the first reinforcement learning algorithm of this kind. Our method specializes to continuous state, continuous action zero-sum Markov games; as such, we provide a provably-convergent policy gradient method for these problems as well. Finally, we prove that our framework naturally models reach-avoid problems, and run experiments which show that the Stackelberg equilibrium policies learned by our method exhibit better safety and liveness properties than their Nash counterparts.

---

[2]To simplify notation, we drop the dependency of action spaces $\mathcal{A}$ and $\mathcal{B}$ on states going forward, but our theory applies in this more general setting.

[3]Unlike $\boldsymbol{a}^{(t)}$ and $\boldsymbol{b}^{(t)}$, which are deterministic actions because they depend on a realized history of states and actions encountered, $A^{(t)}$ and $B^{(t)}$ are random variables, because they might depend on a random history.

## 2 Preliminaries

**Notation.** All notation for variable types, e.g., vectors, should be clear from context; if any confusion arises, see Appendix A. Unless otherwise noted, we assume $\|\cdot\|$ is the Euclidean norm, $\|\cdot\|_2$. We let $\Delta_n = \{\boldsymbol{x} \in \mathbb{R}^n_+ \mid \sum_{i=1}^n x_i = 1\}$ denote the unit simplex in $\mathbb{R}^n$, and $\Delta(A)$, the set of probability distributions on the set $A$. We also define the support of any distribution $f \in \Delta(\mathcal{X})$ as $\mathrm{supp}(f) \doteq \{\boldsymbol{x} \in \mathcal{X} : f(\boldsymbol{x}) > 0\}$. We denote the orthogonal projection operator onto a set $C$ by $\Pi_C$, i.e., $\Pi_C(\boldsymbol{x}) = \arg\min_{\boldsymbol{y} \in C} \|\boldsymbol{x} - \boldsymbol{y}\|^2$. We denote by $\mathbb{1}_\mathcal{C}(\boldsymbol{x})$ the indicator function of a set $\mathcal{C}$, with value 1 if $\boldsymbol{x} \in \mathcal{C}$ and 0 otherwise. Given two vectors $\boldsymbol{x}, \boldsymbol{y} \in \mathbb{R}^n$, we write $\boldsymbol{x} \geq \boldsymbol{y}$ or $\boldsymbol{x} > \boldsymbol{y}$ to mean component-wise $\geq$ or $>$, respectively. For any set $\mathcal{C}$, we denote the diameter by $\mathrm{diam}(\mathcal{C}) \doteq \max_{\boldsymbol{c}, \boldsymbol{c}' \in \mathcal{C}} \|\boldsymbol{c} - \boldsymbol{c}'\|$. Given a tuple consisting of a sequences of iterates and weights $(\{\boldsymbol{z}^{(t)}\}_t, \{\eta^{(t)}\}_t)$, the weighted average of the iterates is given by $\overline{\boldsymbol{z}}_{\boldsymbol{\eta}} \doteq \frac{\sum_t \eta^{(t)} \boldsymbol{z}^{(t)}}{\sum_t \eta^{(t)}}$.

**Mathematical Concepts.** Given $\mathcal{X} \subset \mathbb{R}^n$, the function $f : \mathcal{X} \to \mathcal{Y}$ is said to be $\ell_f$-*Lipschitz-continuous* w.r.t. norm $\|\cdot\|$ iff $\forall \boldsymbol{x}_1, \boldsymbol{x}_2 \in \mathcal{X}, \|f(\boldsymbol{x}_1) - f(\boldsymbol{x}_2)\| \leq \ell_f \|\boldsymbol{x}_1 - \boldsymbol{x}_2\|$. If $\mathcal{Y} = \mathbb{R}$, then $f$ is *convex* (resp. *concave*) iff for all $\lambda \in (0, 1)$ and $\boldsymbol{x}, \boldsymbol{x}' \in \mathcal{X}, f(\lambda \boldsymbol{x} + (1 - \lambda)\boldsymbol{x}') \leq$ (resp. $\geq$) $\lambda f(\boldsymbol{x}) + (1 - \lambda) f(\boldsymbol{x}')$. For any $\mathcal{Y}$, if the relation holds with equality, then $f$ is called *affine*. A two-sided function $h : \mathcal{X} \times \mathcal{Y} \to \mathcal{Z}$ is *biaffine* if $\boldsymbol{x} \mapsto f(\boldsymbol{x}, \boldsymbol{y})$ is affine for all $\boldsymbol{y} \in \mathcal{Y}$, and $\boldsymbol{y} \mapsto h(\boldsymbol{x}, \boldsymbol{y})$ is affine for all $\boldsymbol{x} \in \mathcal{X}$. $f$ is $\mu$-*strongly convex* if $f(\boldsymbol{x}_1) \geq f(\boldsymbol{x}_2) + \langle \nabla_{\boldsymbol{x}} f(\boldsymbol{x}_2), \boldsymbol{x}_1 - \boldsymbol{x}_2 \rangle + \mu/2 \|\boldsymbol{x}_1 - \boldsymbol{x}_1\|^2$. For convenience, we say that an $l$-dimensional vector-valued function $\boldsymbol{g} : \mathcal{X} \to \mathcal{Y} \subset \mathbb{R}^l$ is log-convex, convex, log-concave, or concave, respectively, if $g_k$ is log-convex, convex, log-concave, or concave, for all $k \in [l]$. A correspondence $\mathcal{Z} : \mathcal{X} \to \mathcal{Y}$ is *concave* if for all $\lambda \in (0, 1)$ and $\boldsymbol{x}, \boldsymbol{x}' \in \mathcal{X}$, $\mathcal{Z}(\lambda \boldsymbol{x} + (1 - \lambda)\boldsymbol{x}') \subseteq \lambda \mathcal{Z}(\boldsymbol{x}) + (1 - \lambda)\mathcal{Z}(\boldsymbol{x})$, assuming Minkowski summation [22, 57].

We require notions of stochastic convexity related to stochastic dominance of probability distributions [7]. Given non-empty and convex parameter and outcome spaces $\mathcal{W}$ and $\mathcal{O}$ respectively, a conditional probability distribution $\boldsymbol{w} \mapsto p(\cdot \mid \boldsymbol{w}) \in \Delta(\mathcal{O})$ is said to be *stochastically convex* (resp. *stochastically concave*) in $\boldsymbol{w} \in \mathcal{W}$ if for all continuous, bounded, and convex (resp. concave) functions $v : \mathcal{O} \to \mathbb{R}$, $\lambda \in (0, 1)$, and $\boldsymbol{w}', \boldsymbol{w}^\dagger \in \mathcal{W}$ s.t. $\overline{\boldsymbol{w}} = \lambda \boldsymbol{w}' + (1 - \lambda)\boldsymbol{w}^\dagger$, it holds that $\mathbb{E}_{O \sim p(\cdot \mid \overline{\boldsymbol{w}})}[v(O)] \leq$ (resp $\geq$) $\lambda \mathbb{E}_{O \sim p(\cdot \mid \boldsymbol{w}')}[v(O)] + (1 - \lambda)\mathbb{E}_{O \sim p(\cdot \mid \boldsymbol{w}^\dagger)}[v(O)]$.

## 3 Coupled Min-Max Optimization Problems

A *min-max Stackelberg game*, denoted $(\mathcal{X}, \mathcal{Y}, f, \boldsymbol{g})$, is a two-player, zero-sum game, where one player, called the *leader*, first commits to an action $\boldsymbol{x} \in \mathcal{X}$ from its *action space* $\mathcal{X} \subset \mathbb{R}^n$, after which the second player, called the *follower*, takes an action $\boldsymbol{y} \in \mathcal{Z}(\boldsymbol{x}) \subset \mathcal{Y}$ from a subset of of his *action space* $\mathcal{Y} \subseteq \mathbb{R}^m$ determined by the *action correspondence* $\mathcal{Z} : \mathbb{R}^n \rightrightarrows \mathcal{Y}$. As is standard in the optimization literature, we assume throughout that the follower's action correspondence can be equivalently represented via a *coupling constraint function* $\boldsymbol{g} : \mathbb{R}^n \times \mathbb{R}^m \to \mathbb{R}^d$ s.t. $\mathcal{Z}(\boldsymbol{x}) \doteq \{\boldsymbol{y} \in \mathcal{Y} \mid \boldsymbol{g}(\boldsymbol{x}, \boldsymbol{y}) \geq \boldsymbol{0}\}$. An *action profile* $(\boldsymbol{x}, \boldsymbol{y}) \in \mathcal{X} \times \mathcal{Y}$ comprises actions for both players. Once both players have taken their actions, the leader (resp. follower) receives a loss (resp. payoff) $f(\boldsymbol{x}, \boldsymbol{y})$, defined by an *objective function* $f : \mathbb{R}^n \times \mathbb{R}^m \to \mathbb{R}$. We define the *marginal function* $f^*(\boldsymbol{x}) \doteq \max_{\boldsymbol{y} \in \mathcal{Z}(\boldsymbol{x})} f(\boldsymbol{x}, \boldsymbol{y})$, which, given an action for the leader, outputs its ensuing payoff, assuming the follower best responds. The constraints in a min-max Stackelberg game are said to be *uncoupled* if $\mathcal{Z}(\boldsymbol{x}) = \mathcal{Y}$, for all $\boldsymbol{x} \in \mathcal{X}$. A min-max Stackelberg game is said to be *continuous* iff 1. the objective function $f$ is continuous; 2. the action spaces $\mathcal{X}$ and $\mathcal{Y}$ are non-empty and compact; and 3. the action correspondence $\mathcal{Z}$ is continuous, non-empty- and compact-valued.[4]

**Stackelberg Equilibrium.** The canonical solution concept for min-max Stackelberg games is the $(\varepsilon, \delta)$-*Stackelberg equilibrium* ($(\varepsilon, \delta)$-*SE, or SE if $\varepsilon = \delta = 0$*), an action profile $(\boldsymbol{x}^*, \boldsymbol{y}^*) \in \mathcal{X} \times \mathcal{Y}$ s.t. $\|\Pi_{\mathbb{R}^d}[\boldsymbol{g}(\boldsymbol{x}^*, \boldsymbol{y}^*)]\| \leq \delta$ and $\min_{\boldsymbol{x} \in \mathcal{X}} f^*(\boldsymbol{x}) + \varepsilon \geq f(\boldsymbol{x}^*, \boldsymbol{y}^*) \geq \max_{\boldsymbol{y} \in \mathcal{Z}(\boldsymbol{x}^*)} f(\boldsymbol{x}^*, \boldsymbol{y}) - \delta$, for $\varepsilon, \delta \geq 0$.[5] As a straightforward corollary of Theorem 3.2 of Goktas and Greenwald [33], a SE is guaranteed to exist in continuous Stackelberg games. Moreover, the set of SE can be characterized as solutions to the following *coupled min-max optimization problem*: $\min_{\boldsymbol{x} \in \mathcal{X}} \max_{\boldsymbol{y} \in \mathcal{Z}(\boldsymbol{x})} f(\boldsymbol{x}, \boldsymbol{y})$.

---

[4]See Theorem 5.9 and Example 5.10 of Rockafellar and Wets [63] for conditions on $\boldsymbol{g}$ that guarantee the continuity of $\mathcal{Z}$ or Section 3 of Goktas and Greenwald [33].

[5]For $\delta > 0$, this definition of an $(\varepsilon, \delta)$-SE is more general than the one introduced by Goktas and Greenwald [33], as it allows for the coupling constraints to be satisfied only approximately, which is necessary in this paper, as the coupling constraints can only be accessed via a stochastic oracle.

An alternative but weaker solution concept previously considered for min-max Stackelberg games [71] is the $\varepsilon$-*generalized Nash equilibrium* ($\varepsilon$-GNE, or GNE if $\varepsilon = 0$), i.e., $(\boldsymbol{x}^*, \boldsymbol{y}^*) \in \mathcal{X} \times \mathcal{Z}(\boldsymbol{x}^*)$ s.t. $\min_{\boldsymbol{x} \in \mathcal{X}} f(\boldsymbol{x}, \boldsymbol{y}^*) + \varepsilon \geq f(\boldsymbol{x}^*, \boldsymbol{y}^*) \geq \max_{\boldsymbol{y} \in \mathcal{Z}(\boldsymbol{x}^*)} f(\boldsymbol{x}^*, \boldsymbol{y}) - \varepsilon$, for some $\varepsilon \geq 0$.[6] In general, the set of GNE and SE need not intersect; as such, GNE are not necessarily solutions of $\min_{\boldsymbol{x} \in \mathcal{X}} \max_{\boldsymbol{y} \in \mathcal{Z}(\boldsymbol{x})} f(\boldsymbol{x}, \boldsymbol{y})$ (see, Appendix A of Goktas and Greenwald [33]). Furthermore, there is no GNE whose value is less than the SE value of a game. When a min-max Stackelberg game's constraints are uncoupled, a(n $\varepsilon$-)GNE is called a(n $\varepsilon$-)*saddle point*, or a(n $\varepsilon$-)*Nash equilibrium*, and *is* also an SE. Finally, a saddle point is guaranteed to exist [67, 73] in continuous min-max Stackelberg games with uncoupled constraints, a convex-concave objective $f$, and convex action spaces $\mathcal{X}$ and $\mathcal{Y}$, in which case such games have traditionally been referred to as convex-concave min-max (simultaneous-move) games or saddle-point problems [13].

**Convex-Concave Games.** A min-max Stackelberg game is said to be *convex-concave* if, in addition to being continuous, 1. $f^*$ is convex; 2. $\boldsymbol{y} \mapsto f(\boldsymbol{x}, \boldsymbol{y})$ is concave, for all $\boldsymbol{x} \in \mathcal{X}$; 3. $\mathcal{X}$ and $\mathcal{Y}$ are convex; and 4. $\mathcal{Z}$ is convex-valued. This definition generalizes that of convex-concave min-max (simultaneous-move) game, because in such games, the marginal function $f^*$ is necessarily convex when $f$ is convex, by Danskin's theorem [23]. Assuming access to an exact first-order oracle, an $(\varepsilon, \delta)$-SE of a convex-concave min-max Stackelberg game can be computed in polynomial time when $f$ and $\boldsymbol{g}$ are Lipschitz-continuous [33], while the computation is NP-hard in continuous min-max Stackelberg games, even when $\mathcal{X}$ and $\mathcal{Y}$ are convex, $f$ is convex-concave, and $\boldsymbol{g}$ is affine [47].

All the conditions that define a convex-concave Stackelberg game depend on the game primitives, namely $(\mathcal{X}, \mathcal{Y}, f, \boldsymbol{g})$, and are well-understood (see, for instance Section 5 of Rockafellar and Wets [63]), with the exception of the condition that the marginal function $f^*$ be convex. While it is difficult to obtain necessary and sufficient conditions on the game primitives that ensure the convexity of $f^*$, one possibility is to require $f$ to be convex in $(\boldsymbol{x}, \boldsymbol{y})$ and $\mathcal{Z}$ to be concave.[7] The following sufficient conditions, which also guarantee concavity, were introduced by Goktas and Greenwald [33].

**Assumption 1** (Convex-Concave Assumptions). *1. The objective function $f(\boldsymbol{x}, \boldsymbol{y})$ is convex in $(\boldsymbol{x}, \boldsymbol{y})$, and concave in $\boldsymbol{y}$, for all $\boldsymbol{x} \in \mathcal{X}$; 2. the action correspondence $\mathcal{Z}$ is concave; 3. the action spaces $\mathcal{X}$ and $\mathcal{Y}$ are convex.*

As these assumptions are only sufficient, they are not necessarily satisfied in all applications of convex-concave min-max Stackelberg game. Hence, the convexity of the marginal function must sometimes be established by other means. We thus provide the following alternative set of sufficient conditions, which we use to show that the reach-avoid problem we study in Section 5 is convex-concave.

**Assumption 2** (Alternative Convex-Concave Assumptions). *1. (Convex-concave objective) The objective $f(\boldsymbol{x}, \boldsymbol{y})$ is convex in $\boldsymbol{x}$, for all $\boldsymbol{y} \in \mathcal{Y}$, and concave in $\boldsymbol{y}$, for all $\boldsymbol{x} \in \mathcal{X}$; 2. (log-convex-concave coupling) the coupling constraint $\boldsymbol{g}(\boldsymbol{x}, \boldsymbol{y})$ is log-convex in $\boldsymbol{x}$ for all $\boldsymbol{y} \in \mathcal{Y}$, and concave in $\boldsymbol{y}$ for all $\boldsymbol{x} \in \mathcal{X}$; and 3. the action spaces $\mathcal{X}$ and $\mathcal{Y}$ are convex.*

**Computation.** We now turn our attention to the computation of $(\varepsilon, \delta)$-SE in convex-concave min-max Stackelberg games, assuming access to an *unbiased first-order stochastic oracle* $(\widehat{F}, \widehat{G}, \mathcal{F}, \mathcal{G})$ comprising random functions $\widehat{F} : \mathbb{R}^n \times \mathbb{R}^m \times \Theta \to \mathbb{R}$ and $\widehat{G} : \mathbb{R}^n \times \mathbb{R}^m \times \Phi \to \mathbb{R}^d$ and sampling distributions $\mathcal{F} \in \Delta(\Theta)$ and $\mathcal{G} \in \Delta(\Phi)$ s.t. $\mathbb{E}_{\boldsymbol{\theta} \sim \mathcal{F}}[\widehat{F}(\boldsymbol{x}, \boldsymbol{y}; \boldsymbol{\theta})] = f(\boldsymbol{x}, \boldsymbol{y})$, $\mathbb{E}_{\boldsymbol{\phi} \sim \mathcal{G}}[\widehat{G}(\boldsymbol{x}, \boldsymbol{y}; \boldsymbol{\phi})] = \boldsymbol{g}(\boldsymbol{x}, \boldsymbol{y})$, $\mathbb{E}_{\boldsymbol{\theta}}[\nabla_{(\boldsymbol{x}, \boldsymbol{y})} \widehat{F}(\boldsymbol{x}, \boldsymbol{y}; \boldsymbol{\theta})] = \nabla f(\boldsymbol{x}, \boldsymbol{y})$, and $\mathbb{E}_{\boldsymbol{\phi}}[\nabla_{(\boldsymbol{x}, \boldsymbol{y})} \widehat{G}(\boldsymbol{x}, \boldsymbol{y}; \boldsymbol{\phi})] = \nabla \boldsymbol{g}(\boldsymbol{x}, \boldsymbol{y})$. The following assumptions are required for the convergence of our methods.

**Assumption 3.** *1. (Lipschitz game) $f$ and $\boldsymbol{g}$ are Lipschitz-continuous; 2. (concave representation) the coupling constraint function $\boldsymbol{y} \mapsto \boldsymbol{g}(\boldsymbol{x}, \boldsymbol{y})$ is concave for all $\boldsymbol{x} \in \mathcal{X}$; 3. (Slater's condition) $\forall \boldsymbol{x} \in \mathcal{X}, \exists \widehat{\boldsymbol{y}} \in \mathcal{Y}$ s.t. $\boldsymbol{g}(\boldsymbol{x}, \widehat{\boldsymbol{y}}) > 0$; and 4. (stochastic oracle) there exists an unbiased first-order stochastic oracle $(\widehat{F}, \widehat{G}, \mathcal{F}, \mathcal{G})$ with bounded variance s.t. $\forall (\boldsymbol{x}, \boldsymbol{y}) \in \mathcal{X} \times \mathcal{Y}, \mathbb{E}[\|\widehat{G}(\boldsymbol{x}, \boldsymbol{y}; \boldsymbol{\phi})\|^2] \leq \sigma_{\boldsymbol{g}}$, $\mathbb{E}[\|\nabla_{(\boldsymbol{x}, \boldsymbol{y})} \widehat{F}(\boldsymbol{x}, \boldsymbol{y}; \boldsymbol{\theta})\|^2] \leq \sigma_{\nabla f}$, and $\mathbb{E}[\|\nabla_{(\boldsymbol{x}, \boldsymbol{y})} \widehat{G}(\boldsymbol{x}, \boldsymbol{y}; \boldsymbol{\phi})\|^2] \leq \sigma_{\nabla \boldsymbol{g}}$, for $\sigma_{\boldsymbol{g}}, \sigma_{\nabla f}, \sigma_{\nabla \boldsymbol{g}} \geq 0$.*

In the sequel, we rely on the following notation and definitions. For any action $\boldsymbol{x} \in \mathcal{X}$ of the leader, we can re-express the marginal function in terms of the *Lagrangian* $\ell(\boldsymbol{y}, \boldsymbol{\lambda}; \boldsymbol{x}) \doteq f(\boldsymbol{x}, \boldsymbol{y}) + \langle \boldsymbol{\lambda}, \boldsymbol{g}(\boldsymbol{x}, \boldsymbol{y}) \rangle$ (see, for instance, Section 5 of Boyd et al. [17]) as follows: $f^*(\boldsymbol{x}) = \max_{\boldsymbol{y} \in \mathcal{Y}} \min_{\boldsymbol{\lambda} \in \mathbb{R}_+^d} \ell(\boldsymbol{y}, \boldsymbol{\lambda}; \boldsymbol{x})$. Further, we define the follower's best-response correspondence

---

[6]A GNE is guaranteed to exist in continuous min-max Stackelberg games when the objective function $f$ is convex-concave, the action spaces $\mathcal{A}$ and $\mathcal{B}$ are convex, and the action correspondence $\mathcal{Z}$ is convex-valued [4].

[7]See Section 2 of Nikodem [57] and Chapter 36 of Czerwik [22] for conditions on $\boldsymbol{g}$ which guarantee that $\mathcal{Z}$ is concave and/or continuous and/or convex-valued.

$\mathcal{Y}^*(\boldsymbol{x}) \doteq \arg\max_{\boldsymbol{y}\in\mathcal{Y}} \min_{\boldsymbol{\lambda}\in\mathbb{R}_+^d} \ell(\boldsymbol{y}, \boldsymbol{\lambda}; \boldsymbol{x})$, and the KKT multiplier correspondence $\Lambda^*(\boldsymbol{x}) \doteq \arg\min_{\boldsymbol{\lambda}\in\mathbb{R}_+^d} \max_{\boldsymbol{y}\in\mathcal{Y}} \ell(\boldsymbol{y}, \boldsymbol{\lambda}; \boldsymbol{x})$. With these definitions in hand, under Assumption 3, we can build an unbiased first-order stochastic oracle $\widehat{\mathcal{L}}(\boldsymbol{y}, \boldsymbol{\lambda}; \boldsymbol{x}, \boldsymbol{\theta}, \boldsymbol{\phi}) \doteq \widehat{F}(\boldsymbol{x}, \boldsymbol{y}; \boldsymbol{\theta}) + <\boldsymbol{\lambda}, \widehat{G}(\boldsymbol{x}, \boldsymbol{y}; \boldsymbol{\phi})>$ for the Lagrangian $\ell$ s.t. $\mathbb{E}_{(\boldsymbol{\theta},\boldsymbol{\phi})}[\widehat{\mathcal{L}}(\boldsymbol{y}, \boldsymbol{\lambda}; \boldsymbol{x}, \boldsymbol{\theta}, \boldsymbol{\phi})]$, where the expectation is taken over $(\boldsymbol{\theta}, \boldsymbol{\phi}) \sim \mathcal{F} \times \mathcal{G}$.

**Algorithms.** Assuming access to an exact first-order oracle $(f, \boldsymbol{g})$, a natural approach to computing SE in convex-concave min-max Stackelberg games with uncoupled constraints games (i.e., saddle-point problems) is to simultaneously run projected gradient descent and projected gradient ascent on the objective function $f$ w.r.t. $\boldsymbol{x} \in \mathcal{X}$ and $\boldsymbol{y} \in \mathcal{Y}$, i.e., for $t = 0, 1, 2, \ldots$, $(\boldsymbol{x}^{(t+1)}, \boldsymbol{y}^{(t+1)}) \leftarrow \Pi_{\mathcal{X}\times\mathcal{Y}}[(\boldsymbol{x}^{(t)}, \boldsymbol{y}^{(t)}) + (-\nabla_{\boldsymbol{x}} f, \nabla_{\boldsymbol{y}} f)(\boldsymbol{x}^{(t)}, \boldsymbol{y}^{(t)})]$, a method known under the names of *Arrow-Hurwicz-Uzawa*, *primal-dual*, and (simultaneous) *gradient descent ascent (GDA)* [5, 6]. Intuitively, any fixed point of GDA in such games, i.e., $(\boldsymbol{x}^*, \boldsymbol{y}^*) \in \mathcal{X} \times \mathcal{Y}$ s.t. $\|(\boldsymbol{x}^*, \boldsymbol{y}^*) - \Pi_{\mathcal{X}\times\mathcal{Y}}[(\boldsymbol{x}^*, \boldsymbol{y}^*) + (-\nabla_{\boldsymbol{x}} f, \nabla_{\boldsymbol{y}} f)(\boldsymbol{x}^*, \boldsymbol{y}^*)]\| = 0$, satisfies the necessary and sufficient optimality condition for an action profile to be a SE. More generally, in convex-concave min-max Stackelberg games (with*out* coupled constraints), this approach fails, as the necessary and sufficient optimality condition for an action profile $(\boldsymbol{x}^*, \boldsymbol{y}^*) \in \mathcal{X} \times \mathcal{Y}$ to be a SE is $\|(\boldsymbol{x}^*, \boldsymbol{y}^*) - \Pi_{\mathcal{X}\times\mathcal{Z}(\boldsymbol{x}^*)}[(\boldsymbol{x}^*, \boldsymbol{y}^*) + (-\nabla_{\boldsymbol{x}} f^*(\boldsymbol{x}^*), \nabla_{\boldsymbol{y}} f(\boldsymbol{x}^*, \boldsymbol{y}^*))]\| = 0$, where, for any leader action $\widehat{\boldsymbol{x}} \in \mathcal{X}$, $\nabla_{\boldsymbol{x}} f^*(\widehat{\boldsymbol{x}}) \doteq \ell(\boldsymbol{y}^*(\widehat{\boldsymbol{x}}), \boldsymbol{\lambda}^*(\widehat{\boldsymbol{x}}); \widehat{\boldsymbol{x}})$, for some $(\boldsymbol{y}^*, \boldsymbol{\lambda}^*)(\widehat{\boldsymbol{x}}) \in \mathcal{Y}^*(\widehat{\boldsymbol{x}}) \times \Lambda^*(\widehat{\boldsymbol{x}})$, by the subdifferential envelope theorem [33]. The observation that any subgradient of $\nabla_{\boldsymbol{x}} f^*$ depends on the optimal KKT multipliers motivates a first-order method based on the gradient of the Lagrangian.

A min-max Stackelberg game can be seen as a three-player game $\min_{\boldsymbol{x}\in\mathcal{X}} \max_{\boldsymbol{y}\in\mathcal{Z}(\boldsymbol{x})} f(\boldsymbol{x}, \boldsymbol{y}) = \min_{\boldsymbol{x}\in\mathcal{X}} \max_{\boldsymbol{y}\in\mathcal{Y}} \min_{\boldsymbol{\lambda}\in\mathbb{R}_+^d} \ell(\boldsymbol{y}, \boldsymbol{\lambda}; \boldsymbol{x})$, where the $\boldsymbol{x}$-player moves first, and the $\boldsymbol{y}$- and $\boldsymbol{\lambda}$-players move second, simultaneously, because strong duality holds under Assumption 3 (Slater's condition [68]) for the inner min-max optimization problem, i.e., $\max_{\boldsymbol{y}\in\mathcal{Y}} \min_{\boldsymbol{\lambda}\in\mathbb{R}_+^d} \ell(\boldsymbol{y}, \boldsymbol{\lambda}; \boldsymbol{x}) = \min_{\boldsymbol{\lambda}\in\mathbb{R}_+^d} \max_{\boldsymbol{y}\in\mathcal{Y}} \ell(\boldsymbol{y}, \boldsymbol{\lambda}; \boldsymbol{x})$. The problem of computing an SE can thus be reduced to the min-max optimization $\min_{(\boldsymbol{x},\boldsymbol{\lambda})\in\mathcal{X}\times\mathbb{R}_+^d} \max_{\boldsymbol{y}\in\mathcal{Y}} \ell(\boldsymbol{y}, \boldsymbol{\lambda}; \boldsymbol{x})$, which we might hope to solve by running GDA on $\ell(\boldsymbol{y}, \boldsymbol{\lambda}; \boldsymbol{x})$ w.r.t. $(\boldsymbol{x}, \boldsymbol{\lambda})$ and $\boldsymbol{y}$ over $\mathcal{X} \times \mathbb{R}_+^d$ and $\mathcal{Y}$, respectively. Although $\boldsymbol{y} \mapsto \ell(\boldsymbol{y}, \boldsymbol{\lambda}; \boldsymbol{x})$ is concave, $(\boldsymbol{x}, \boldsymbol{\lambda}) \mapsto \ell(\boldsymbol{y}, \boldsymbol{\lambda}; \boldsymbol{x})$ is not convex, and its stationary points (i.e., points $(\boldsymbol{y}^*, \boldsymbol{\lambda}^*; \boldsymbol{x}^*)$ s.t. $\|(\boldsymbol{y}^*, \boldsymbol{\lambda}^*; \boldsymbol{x}^*) - \Pi_{\mathcal{Y}\times\mathbb{R}_+^d\times\mathcal{X}}[(\boldsymbol{y}^*, \boldsymbol{\lambda}^*; \boldsymbol{x}^*) + (\nabla_{\boldsymbol{y}}\ell, -\nabla_{\boldsymbol{\lambda}}\ell, -\nabla_{\boldsymbol{x}}\ell)(\boldsymbol{y}^*, \boldsymbol{\lambda}^*; \boldsymbol{x}^*)]\| = 0$) do not necessarily coincide with SE even in simple convex-concave min-max Stackelberg games [35].

---

**Algorithm 1** Saddle-Point-Oracle SGD/Nested SGDA

**Inputs:** $\mathcal{X}, \mathcal{Y}, \widehat{F}, \widehat{G}, \mathcal{F}, \mathcal{G}, \boldsymbol{x}^{(0)}, T_{\boldsymbol{x}}, \{\eta_{\boldsymbol{x}}^{(t)}\}_t, \delta$
(+ for nested SGDA:) $\Lambda, \boldsymbol{y}'^{(0)}, \boldsymbol{\lambda}'^{(0)}, T_{\boldsymbol{y}}, \{\eta_{\boldsymbol{y}}^{(t)}\}_t$
**Outputs:** $(\boldsymbol{x}^{(t)}, \boldsymbol{y}^{(t)}, \boldsymbol{\lambda}^{(t)})_{t=0}^{T_{\boldsymbol{x}}}$

1: **for** $t = 0, \ldots, T_{\boldsymbol{x}}$ **do**
2:    **if** *Saddle-Point-Oracle SGD* **then**
3:       Find $(\boldsymbol{y}^{(t)}, \boldsymbol{\lambda}^{(t)}) \in \mathcal{Y} \times \mathbb{R}_+^d$ s.t.
4:       $\max_{\boldsymbol{y}\in\mathcal{Y}} \ell(\boldsymbol{y}^{(t)}, \boldsymbol{\lambda}; \boldsymbol{x}^{(t)}) - \min_{\boldsymbol{\lambda}\in\mathbb{R}_+^d} \ell(\boldsymbol{y}, \boldsymbol{\lambda}^{(t)}; \boldsymbol{x}^{(t)}) \leq \delta$,
5:    **if** *Nested SGDA* **then**
6:       **for** $s = 0, \ldots, T_{\boldsymbol{y}}$ **do**
7:          Sample $\boldsymbol{\theta} \sim \mathcal{F}, \boldsymbol{\phi} \sim \mathcal{G}$
8:          $\boldsymbol{y}'^{(s+1)} \leftarrow \Pi_{\mathcal{Y}}\Big[\boldsymbol{y}'^{(s)} + \eta_{\boldsymbol{y}}^{(s)} \nabla_{\boldsymbol{y}} \widehat{\mathcal{L}}(\boldsymbol{y}'^{(s)}, \boldsymbol{\lambda}'^{(s)}; \boldsymbol{x}^{(t)}, \boldsymbol{\theta}, \boldsymbol{\phi})\Big]$
9:          $\boldsymbol{\lambda}'^{(s+1)} \leftarrow \Pi_{\Lambda}\Big[\boldsymbol{\lambda}'^{(s)} - \eta_{\boldsymbol{y}}^{(s)} \nabla_{\boldsymbol{\lambda}} \widehat{\mathcal{L}}(\boldsymbol{y}'^{(s)}, \boldsymbol{\lambda}'^{(s)}; \boldsymbol{x}^{(t)}, \boldsymbol{\theta}, \boldsymbol{\phi})\Big]$
10:       Average iterates $(\boldsymbol{y}^{(t)}, \boldsymbol{\lambda}^{(t)}) \leftarrow (\overline{\boldsymbol{y}}'_{\eta_{\boldsymbol{y}}}, \overline{\boldsymbol{\lambda}}'_{\eta_{\boldsymbol{y}}})$
11:    Sample $\boldsymbol{\theta} \sim \mathcal{F}, \boldsymbol{\phi} \sim \mathcal{G}$
12:    $\boldsymbol{x}^{(t+1)} \leftarrow \Pi_{\mathcal{X}}\Big[\boldsymbol{x}^{(t)} - \eta_{\boldsymbol{x}}^{(t)} \nabla_{\boldsymbol{x}} \widehat{\mathcal{L}}(\boldsymbol{y}^{(t)}, \boldsymbol{\lambda}^{(t)}; \boldsymbol{x}^{(t)}, \boldsymbol{\theta}, \boldsymbol{\phi})\Big]$
13: **return** $(\overline{\boldsymbol{x}}_{\eta_{\boldsymbol{x}}}, \boldsymbol{y}^{(T_{\boldsymbol{x}})}, \boldsymbol{\lambda}^{(T_{\boldsymbol{x}})})$

---

As GDA fails in these games, Goktas and Greenwald [33] developed *nested GDA*, a nested first-order method for computing an $(\varepsilon, \delta)$-SE, which solves the inner maximization problem by running GDA on $\ell$ w.r.t. $\boldsymbol{y}$ and $\boldsymbol{\lambda}$ over constraint sets $\mathcal{Y}$ and $\mathbb{R}_+^d$ until convergence to a $\delta$-saddle point $(\widehat{\boldsymbol{y}}, \widehat{\boldsymbol{\lambda}})$. Then, exploiting the convexity of the marginal function $f^*$, the algorithm runs a descent step on $f^*$ w.r.t. $\boldsymbol{x}$, in which, for any leader action $\boldsymbol{x} \in \mathcal{X}$, a subgradient $\nabla_{\boldsymbol{x}} f^*$ is approximated by $\widehat{\nabla_{\boldsymbol{x}} f^*}(\boldsymbol{x}) = \ell(\widehat{\boldsymbol{y}}, \widehat{\boldsymbol{\lambda}}; \boldsymbol{x})$. In this paper, we replace the exact first-order oracle used by nested GDA with a stochastic one, the gradient descent step with a step of stochastic gradient descent (SGD), and GDA with stochastic GDA (SGDA), using in both cases the stochastic Lagrangian oracle $\widehat{\mathcal{L}}$. We call our method nested stochastc gradient descent ascent (nested SGDA).

We begin by presenting *saddle-point-oracle stochastic gradient descent algorithm* (saddle-point-oracle SGD, Algorithm 1), whose analysis we build on to develop our primary contribution, nested SGDA. Following Goktas and Greenwald's [33] max-oracle gradient descent algorithm, saddle-point-oracle SGD runs SGD on $f^*$, assuming access to an oracle, which, for any leader action $\boldsymbol{x} \in \mathcal{X}$, returns a $\delta$-saddle point of $(\boldsymbol{y}, \boldsymbol{\lambda}) \mapsto \ell(\boldsymbol{y}, \boldsymbol{\lambda}; \boldsymbol{x})$. Our second algorithm, *nested stochastic gradient descent ascent* (nested SGDA, Algorithm 1), follows the same logic as saddle-point-oracle SGD, but implements the saddle-point oracle by running SGDA. The following theorem establishes conditions under which both of our algorithms converge to an $(\varepsilon + \delta, \delta)$-SE in a number of oracle calls that is polynomial in $1/\varepsilon$ and $1/\delta$.[8]

**Theorem 3.1.** *Let $(\mathcal{X}, \mathcal{Y}, f, \boldsymbol{g})$ be a convex-concave min-max Stackelberg game for which Assumption 3 holds. For any $\varepsilon, \delta \geq 0$, if nested SGDA (resp. saddle-point-oracle SGD) is run with inputs[9] that satisfy for all $t \in \mathbb{N}_+$, $\eta_{\boldsymbol{x}}^{(t)}, \eta_{\boldsymbol{x}}^{(t)} \in \Theta\left(1/\sqrt{t+1}\right)$, and outputs $(\boldsymbol{x}^*, \boldsymbol{y}^*, \boldsymbol{\lambda}^*)$, then in expectation over all runs of the algorithm (i.e., sample paths of $\boldsymbol{\theta}$ and $\boldsymbol{\phi}$), the action profile $(\boldsymbol{x}^*, \boldsymbol{y}^*)$ is an $(\varepsilon + \delta, \delta)$-SE after $\tilde{O}(1/\varepsilon^2\delta^2)$ (resp. $\tilde{O}(1/\varepsilon^2)$) oracle calls. If, in addition, $f^*$ is $\mu$-strongly-convex, then $(\boldsymbol{x}^*, \boldsymbol{y}^*)$ is an $(\varepsilon + \delta, \delta)$-SE after $\tilde{O}(1/\varepsilon\delta^2)$ (resp. $\tilde{O}(1/\varepsilon)$) oracle calls.*

# 4 Policy Gradient in Convex-Concave Zero-Sum Markov Stackelberg Games

In this section, we reduce the computation of Stackelberg equilibrium in zero-sum Markov Stackelberg games to a coupled min-max optimization problem, which enables us to derive a policy gradient method for these games based on our nested SGDA algorithm.

We consider zero-sum Markov Stackelberg games $\mathcal{M} \doteq (l, n, m, d, \mathcal{S}, \mathcal{A}, \mathcal{B}, \mu, r, \boldsymbol{g}, p, \gamma)$ with state space $\mathcal{S} \subset \mathbb{R}^l$ and action spaces $\mathcal{A} \subset \mathbb{R}^n$ and $\mathcal{B} \subset \mathbb{R}^m$ for the leader and follower, respectively, where the follower's actions are constrained by the leader's via vector-valued state-dependent coupling constraints $\boldsymbol{g} : \mathcal{S} \times \mathbb{R}^n \times \mathbb{R}^m \to \mathbb{R}^d$ s.t. that define a correspondence $\mathcal{C}(\boldsymbol{s}, \boldsymbol{a}) \doteq \{\boldsymbol{b} \in \mathcal{B} \mid \boldsymbol{g}(\boldsymbol{s}, \boldsymbol{a}, \boldsymbol{b}) \geq \boldsymbol{0}\}$. We define the set of states with non-trivially coupled constraints $\mathcal{N} \doteq \{\boldsymbol{s} \in \mathcal{S} \mid \exists(\boldsymbol{a}, \boldsymbol{b}) \in \mathcal{A} \times \mathcal{B}, \boldsymbol{g}(\boldsymbol{s}, \boldsymbol{a}, \boldsymbol{b}) < \boldsymbol{0}\}$. A *Markov* policy for the leader (resp. follower)—hereafter policy for short—is one that is history independent, and thus a mapping from states to actions $\boldsymbol{\pi_a} : \mathcal{S} \to \mathcal{A}$ (resp. $\boldsymbol{\pi_b} : \mathcal{S} \to \mathcal{B}$). A *policy profile* $\boldsymbol{\pi} \doteq (\boldsymbol{\pi_a}, \boldsymbol{\pi_b}) \in \mathcal{A}^{\mathcal{S}} \times \mathcal{B}^{\mathcal{S}}$ is a tuple comprising policies for the leader and follower, respectively. The follower's feasible policy correspondence is given by $\mathcal{Z}(\boldsymbol{\pi_a}) = \{\boldsymbol{\pi_b} : \mathcal{S} \to \mathcal{B} \mid \forall \boldsymbol{s} \in \mathcal{N}, \boldsymbol{g}(\boldsymbol{s}, \boldsymbol{\pi}(\boldsymbol{s})) \geq \boldsymbol{0}\}$.

A continuous *action* zero-sum Markov Stackelberg game is a game where 1. for all states $\boldsymbol{s} \in \mathcal{S}$, the reward function $(\boldsymbol{a}, \boldsymbol{b}) \mapsto r(\boldsymbol{s}, \boldsymbol{a}, \boldsymbol{b})$ is continuous and bounded, i.e., $\|r(\boldsymbol{s}, \cdot, \cdot)\|_\infty \leq r_{\max} < \infty$, for some $r_{\max} \in \mathbb{R}_+$; 2. the action spaces $\mathcal{A}$ and $\mathcal{B}$ are non-empty and compact; and 3. for all states $\boldsymbol{s} \in \mathcal{S}$, the correspondence $\boldsymbol{a} \mapsto \mathcal{C}(\boldsymbol{s}, \boldsymbol{a})$ is continuous, non-empty-, and compact-valued. A continuous *state* zero-sum Markov Stackelberg game is a game where 1. $\mathcal{S}$ is non-empty and compact and 2. the reward function $r$ is continuous and bounded, i.e., $\|r\|_\infty < \infty$.

A *history* $\boldsymbol{h} \in (\mathcal{S} \times \mathcal{A} \times \mathcal{B})^\tau$ of length $\tau \in \mathbb{N}$ is a sequence of state-action tuples $\boldsymbol{h} = (\boldsymbol{s}^{(t)}, \boldsymbol{a}^{(t)}, \boldsymbol{b}^{(t)})_{t=0}^{\tau-1}$. Given a policy profile $\boldsymbol{\pi}$ and a history of play $\boldsymbol{h}$ of length $\tau$, we define the *discounted history distribution* as $\nu^{\boldsymbol{\pi}, \tau}(\boldsymbol{h}) = \mu(\boldsymbol{s}^{(0)}) \prod_{t=0}^{\tau-1} \gamma^t p(\boldsymbol{s}^{(t+1)} \mid \boldsymbol{s}^{(t)}, \boldsymbol{a}^{(t)}, \boldsymbol{b}^{(t)}) \mathbb{1}_{\boldsymbol{\pi}(\boldsymbol{s}^{(t)})}(\boldsymbol{a}^{(t)}, \boldsymbol{b}^{(t)})$. Define the set of all realizable trajectories $\mathcal{H}^{\boldsymbol{\pi}, \tau}$ of length $\tau$ under policy $\boldsymbol{\pi}$ as $\mathcal{H}^{\boldsymbol{\pi}, \tau} \doteq \text{supp}(\nu^{\boldsymbol{\pi}, \tau})$, i.e., the set of all histories that occur with non-zero probability. For convenience, we denote by $\nu^{\boldsymbol{\pi}} \doteq \nu^{\boldsymbol{\pi}, \infty}$, and by $H = (S^{(t)}, A^{(t)}, B^{(t)})_t$ any randomly sampled history from $\nu^{\boldsymbol{\pi}}$. Finally, we define the *discounted state-visitation distribution* under any initial state distribution $\mu$ as $\delta_\mu^{\boldsymbol{\pi}}(\boldsymbol{s}) = \sum_{t=0}^\infty \sum_{\boldsymbol{h} \in \mathcal{H}^{\boldsymbol{\pi}, t} : S^{(t)} = \boldsymbol{s}} \mu(\boldsymbol{s}^{(0)}) \prod_{k=1}^t \gamma^k p(\boldsymbol{s}^{(k)} \mid \boldsymbol{s}^{(k-1)}, \boldsymbol{a}^{(k-1)}, \boldsymbol{b}^{(k-1)})$.

Given a policy profile $\boldsymbol{\pi}$, the *(state-)value function* $v^{\boldsymbol{\pi}} : \mathcal{S} \to \mathbb{R}$ and the *action-value function* $q^{\boldsymbol{\pi}} : \mathcal{S} \times \mathcal{A} \times \mathcal{B} \to \mathbb{R}$ are defined in terms of $\nu^{\boldsymbol{\pi}}$ as $v^{\boldsymbol{\pi}}(\boldsymbol{s}) \doteq \mathbb{E}_{H \sim \nu^{\boldsymbol{\pi}}}\left[\sum_{t=0}^\infty r\left(S^{(t)}, A^{(t)}, B^{(t)}\right) \mid S^{(0)} = \boldsymbol{s}\right]$ and $q^{\boldsymbol{\pi}}(\boldsymbol{s}, \boldsymbol{a}, \boldsymbol{b}) \doteq \mathbb{E}_{H \sim \nu^{\boldsymbol{\pi}}}\left[\sum_{t=0}^\infty r\left(S^{(t)}, A^{(t)}, B^{(t)}\right) \mid S^{(0)} = \boldsymbol{s}, A^{(0)} = \boldsymbol{a}, B^{(0)} = \boldsymbol{b}\right]$, respectively. The *cumulative payoff function* $u : \mathcal{A}^{\mathcal{S}} \times \mathcal{B}^{\mathcal{S}} \to \mathbb{R}$ is then defined in terms of the value function as $u(\boldsymbol{\pi_a}, \boldsymbol{\pi_b}) \doteq \mathbb{E}_{S \sim \mu}[v^{\boldsymbol{\pi}}(S)]$, i.e., the total expected loss (resp. gain) of the leader (resp.

---

[8]We include detailed theorem statements and proofs in the full version of the paper.

[9]$\Lambda$ should be chosen as a superset of all optimal KKT multipliers, i.e., $\cup_{\boldsymbol{x} \in \mathcal{X}} \Lambda^*(\boldsymbol{x}) \subseteq \Lambda$ (see Appendix C).

follower). Additionally, the *marginal action-value function* $q^{*\boldsymbol{\pi}}(\boldsymbol{s}, \boldsymbol{a}) \doteq \max_{\boldsymbol{b} \in \mathcal{C}(\boldsymbol{s}, \boldsymbol{a})} q^{\boldsymbol{\pi}}(\boldsymbol{s}, \boldsymbol{a}, \boldsymbol{b})$ is the payoff when play initiates at state $\boldsymbol{s}$ with the leader taking action $\boldsymbol{a}$, after which the follower best responds (at state $\boldsymbol{s}$ only), with both players playing according to $\boldsymbol{\pi}$ thereafter. Finally, for any leader policy $\boldsymbol{\pi_a} \in \mathcal{A}^{\mathcal{S}}$, we define the *marginal (state-value) function* $u^*(\boldsymbol{\pi_a}) \doteq \max_{\boldsymbol{\pi_b} \in \mathcal{Z}(\boldsymbol{\pi_a})} u(\boldsymbol{\pi_a}, \boldsymbol{\pi_b})$.

**Recursive Stackelberg Equilibrium.** A policy profile $\boldsymbol{\pi}^* \doteq (\boldsymbol{\pi_a^*}, \boldsymbol{\pi_b^*}) \in \mathcal{A}^{\mathcal{S}} \times \mathcal{B}^{\mathcal{S}}$ is called an $(\varepsilon, \delta)$-*recursive (or Markov perfect) Stackelberg equilibrium* iff $\forall \boldsymbol{s} \in \mathcal{S}$, $\|\Pi_{\mathbb{R}^d}[\boldsymbol{g}(\boldsymbol{s}, \boldsymbol{\pi}^*(\boldsymbol{s}))]\| \leq \delta$ and $\max_{\boldsymbol{\pi_b} \in \mathcal{Z}(\boldsymbol{\pi_a})} v^{(\boldsymbol{\pi_a^*}, \boldsymbol{\pi_b})}(\boldsymbol{s}) - \delta \leq v^{\boldsymbol{\pi}^*}(\boldsymbol{s}) \leq \min_{\boldsymbol{\pi_a} \in \mathcal{A}^{\mathcal{S}}} \max_{\boldsymbol{\pi_b} \in \mathcal{Z}(\boldsymbol{\pi_a})} v^{(\boldsymbol{\pi_a}, \boldsymbol{\pi_b})}(\boldsymbol{s}) + \varepsilon$. A recursive SE is guaranteed to exist in continuous state, continuous action zero-sum Markov Stackelberg games [36]. A policy profile $\boldsymbol{\pi}^* \doteq (\boldsymbol{\pi_a^*}, \boldsymbol{\pi_b^*}) \in \mathcal{A}^{\mathcal{S}} \times \mathcal{Z}(\boldsymbol{\pi_a^*})$ is called an $(\varepsilon, \delta)$-*Markov perfect GNE* iff $\forall \boldsymbol{s} \in \mathcal{S}$, $\max_{\boldsymbol{\pi_b} \in \mathcal{Z}(\boldsymbol{\pi_a})} v^{(\boldsymbol{\pi_a^*}, \boldsymbol{\pi_b})}(\boldsymbol{s}) - \delta \leq v^{\boldsymbol{\pi}^*}(\boldsymbol{s}) \leq \min_{\boldsymbol{\pi_a} \in \mathcal{A}^{\mathcal{S}}} v^{\boldsymbol{\pi_a}, \boldsymbol{\pi_b^*}}(\boldsymbol{s}) + \varepsilon$.

**Convex-Concave Markov Stackelberg Games.** As we have shown (Theorem 3.1), Stackelberg equilibria can be computed in polynomial time in convex-concave min-max Stackelberg games, assuming access to an unbiased first order-stochastic oracle. We now define an analogous class of Markov Stackelberg games, namely zero-sum Markov Stackelberg games in which the min-max Stackelberg game played at each state is convex-concave. A *convex-concave zero-sum Markov Stackelberg game* is a continuous state, continuous action zero-sum Markov game where, for all policy profiles $\boldsymbol{\pi} \in \mathcal{A}^{\mathcal{S}} \times \mathcal{B}^{\mathcal{S}}$, 1. the marginal action-value function $(\boldsymbol{s}, \boldsymbol{a}) \mapsto q^{*\boldsymbol{\pi}}(\boldsymbol{s}, \boldsymbol{a})$ is convex, 2. the action-value function $(\boldsymbol{s}, \boldsymbol{b}) \mapsto q^{\boldsymbol{\pi}}(\boldsymbol{s}, \boldsymbol{a}, \boldsymbol{b})$ is concave, for all $\boldsymbol{a} \in \mathcal{A}$, 3. the state and action spaces $\mathcal{S}, \mathcal{A}$ and $\mathcal{B}$ are convex, and 4. the action correspondence $\mathcal{C}$ is convex-valued. We note that any *continuous state, continuous action convex-concave zero-sum Markov game*, i.e., 1. $\mathcal{N} = \emptyset$, 2. $(\boldsymbol{s}, \boldsymbol{a}) \mapsto r(\boldsymbol{s}, \boldsymbol{a}, \boldsymbol{b})$ is convex, for all $\boldsymbol{b} \in \mathcal{B}$, 3. $(\boldsymbol{s}, \boldsymbol{b}) \mapsto r(\boldsymbol{s}, \boldsymbol{a}, \boldsymbol{b})$ is concave, for all $\boldsymbol{a} \in \mathcal{A}$, 4. $(\boldsymbol{s}, \boldsymbol{a}) \mapsto p(\cdot \mid \boldsymbol{s}, \boldsymbol{a}, \boldsymbol{b})$ is stochastically convex, for all $\boldsymbol{b} \in \mathcal{B}$; and 5. $(\boldsymbol{s}, \boldsymbol{b}) \mapsto p(\cdot \mid \boldsymbol{s}, \boldsymbol{a}, \boldsymbol{b})$ is stochastically concave, for all $\boldsymbol{a} \in \mathcal{A}$, is a convex-concave zero-sum Markov Stackelberg game for which the set of Markov perfect generalized Nash equilibria is a subset of the recursive SE.

As our plan is to use our nested SGDA algorithm to compute recursive Stackelberg equilibria, we begin by showing that zero-sum Markov Stackelberg games are an instance of min-max Stackelberg games. Assume parametric policy classes for the leader and follower, respectively, namely $\mathcal{P}_{\mathcal{X}} \doteq \{\boldsymbol{\pi_x} : \mathcal{S} \to \mathcal{A} \mid \boldsymbol{x} \in \mathcal{X}\} \subseteq \mathcal{A}^{\mathcal{S}}$ and $\mathcal{P}_{\mathcal{Y}} \doteq \{\boldsymbol{\pi_y} : \mathcal{S} \to \mathcal{B} \mid \boldsymbol{y} \in \mathcal{Y}\} \subseteq \mathcal{B}^{\mathcal{S}}$, for parameter spaces $\mathcal{X} \subset \mathbb{R}^d$ and $\mathcal{Y} \subset \mathbb{R}^d$. Using these parameterizations, we redefine $v^{(\boldsymbol{x}, \boldsymbol{y})} \doteq v^{(\boldsymbol{\pi_x}, \boldsymbol{\pi_y})}$, $q^{(\boldsymbol{x}, \boldsymbol{y})} \doteq q^{(\boldsymbol{\pi_x}, \boldsymbol{\pi_y})}$, $u(\boldsymbol{x}, \boldsymbol{y}) \doteq u(\boldsymbol{\pi_x}, \boldsymbol{\pi_y})$, etc., and thus restate the problem of computing a recursive SE as finding $(\boldsymbol{x}, \boldsymbol{y}) \in \mathcal{X} \times \mathcal{Y}$ that solves $\min_{\boldsymbol{x} \in \mathcal{X}} \max_{\boldsymbol{y} \in \mathcal{Y}: \mathcal{Z}(\boldsymbol{x})} v^{(\boldsymbol{x}, \boldsymbol{y})}(\boldsymbol{s})$, for all states $\boldsymbol{s} \in \mathcal{S}$. As this optimization problem is infinite dimensional for continuous state games, we optimize the objective and satisfy the constraints, both in expectation over the initial state distribution $\mu$, thereby reducing the problem to the min-max Stackelberg game $\min_{\boldsymbol{x} \in \mathcal{X}} \max_{\boldsymbol{y} \in \mathcal{Z}(\boldsymbol{x})} u(\boldsymbol{x}, \boldsymbol{y})$.

In Appendix D, assuming 1. biaffine parametric policy classes, i.e., $(\boldsymbol{s}, \boldsymbol{x}) \mapsto \boldsymbol{\pi_x}(\boldsymbol{s})$ and $(\boldsymbol{s}, \boldsymbol{y}) \mapsto \boldsymbol{\pi_y}(\boldsymbol{s})$ are biaffine, and 2. non-empty, compact, and convex parameter spaces $\mathcal{X}$ and $\mathcal{Y}$, we show that the min-max Stackelberg game associated with any convex-concave zero-sum Markov Stackelberg game is also convex-concave (Lemma 4). We also provide sufficient conditions on the primitives $\mathcal{M}$ of any zero-sum Markov Stackelberg game to ensure that it is convex-concave (Lemma 5 and 6). At a high level, our results allow us to conclude that a zero-sum Markov Stackelberg game is convex-concave if the 1. reward (resp. transition probability) function is concave (resp. stochastically concave) in the state and the follower's action; 2. the reward (resp. transition probability) function is convex (resp. stochastically convex) in the state and the leader's follower's actions; and 3. the follower's action correspondence is concave.

**Computation.** We now turn our attention to the computation of recursive SE in convex-concave zero-sum Markov Stackelberg games. Mirroring the steps by which policy gradient has been show to converge in other settings [24], we first define an unbiased first-order stochastic oracle for zero-sum Markov-Stackelberg games, given access to an unbiased first-order stochastic oracle for the reward and probability transition functions. We then establish convergence of nested SGDA in this setting by invoking Theorem 3.1 under the following assumptions.

**Assumption 4** (Convergence Assumptions). *1. The parameter spaces $\mathcal{X}$ and $\mathcal{Y}$ are non-empty, compact, and convex; 2. the policy parameterizations are biaffine, i.e., $(\boldsymbol{s}, \boldsymbol{x}) \mapsto \boldsymbol{\pi_x}(\boldsymbol{s})$ and $(\boldsymbol{s}, \boldsymbol{y}) \mapsto \boldsymbol{\pi_y}(\boldsymbol{s})$ are biaffine; 3. the set of non-trivially constrained sets is finite $\mathcal{N}$, i.e. $\|\mathcal{N}\| < \infty$;*

*4. (Slater's condition) for all $s \in \mathcal{N}$ and $a \in \mathcal{A}$, there exists $\widehat{b} \in \mathcal{B}$ s.t. $g(s, a, \widehat{b}) > 0$; and 5. the reward $r$, probability transition $p$, and coupling constraint $g$ functions are Lipschitz-continuous.*

Stochastic nested GDA relies on an unbiased first-order stochastic oracle $(\widehat{F}, \widehat{G}, \mathcal{F}, \mathcal{G})$, which we can use to obtain unbiased first-order stochastic estimators of $u$ and $g$. Since the constraints are deterministic, we simply set $\widehat{G}(\boldsymbol{x}, \boldsymbol{y}; \boldsymbol{s}) \doteq (g(s, \boldsymbol{\pi_x}(s), \boldsymbol{\pi_y}(s)))_{s \in \mathcal{S}}$ and $\mathcal{G}(s) \doteq \rho(s)$, for any distribution $\rho \in \Delta(\mathcal{S})$ over the state space to obtain an unbiased first-order stochastic oracle for the constraints $g$. While for simplicity we define $\widehat{G}$ as such, $\widehat{G}$ is tractable to compute (i.e., finite-dimensional) only when $\mathcal{N}$ is finite. When $\mathcal{N}$ is infinite, our theoretical results generalize by setting $\widehat{G}(\boldsymbol{x}, \boldsymbol{y}; \boldsymbol{s}) \doteq (\min_{s \in \mathcal{N}} g_c(s, \boldsymbol{\pi_x}(s), \boldsymbol{\pi_y}(s)))_{c \in [d]}$; however, in practice, this estimator might be intractable, in which case one might choose to abandon our theoretical guarantees in favor of the biased estimator $\widehat{G}(\boldsymbol{x}, \boldsymbol{y}; \boldsymbol{s}) \doteq g(s, \boldsymbol{\pi_x}(s), \boldsymbol{\pi_y}(s))$. In all cases, the definition of $\nabla_{(\boldsymbol{x}, \boldsymbol{y})} \widehat{G}$ follows directly, since $\widehat{G}$ is deterministic. Now, for any history $\boldsymbol{h}$ of length $\tau$, define the *cumulative payoff estimator* $\widehat{R}(\boldsymbol{\pi}; \boldsymbol{h}) \doteq \sum_{t=0}^{\tau-1} \mu(s^{(0)}) \prod_{k=0}^{t-1} \gamma^k p(s^{(k+1)} \mid s^{(k)}, \boldsymbol{\pi}(s^{(k)})) r(s^{(k)}, \boldsymbol{\pi}(s^{(k)})))$. We then construct an estimator for $u$ using *first-order gradient estimator* [69], i.e., we set $\widehat{F}(\boldsymbol{x}, \boldsymbol{y}; \boldsymbol{h}) \doteq \widehat{R}(\boldsymbol{\pi_x}, \boldsymbol{\pi_y}; \boldsymbol{h})$, and $\nabla_{(\boldsymbol{x}, \boldsymbol{y})} \widehat{F}(\boldsymbol{x}, \boldsymbol{y}; \boldsymbol{h}) \doteq \nabla_{(\boldsymbol{x}, \boldsymbol{y})} \widehat{R}(\boldsymbol{\pi_x}, \boldsymbol{\pi_y}; \boldsymbol{h})$. Regarding the variances of this oracle model, as $\widehat{G}$ and $\nabla_{(\boldsymbol{x}, \boldsymbol{y})} \widehat{G}$ are deterministic, they have bounded variance. Moreover, if the policy and the reward and transition probability functions are Lipschitz-continuous, then $\widehat{R}$ and $\nabla_{(\boldsymbol{x}, \boldsymbol{y})} \widehat{R}$ are also Lipschitz-continuous if their domains are compact (i.e., if $\mathcal{S}$, $\mathcal{A}$, and $\mathcal{B}$ are compact). Hence $\widehat{F}$ and $\nabla \widehat{F}$ likewise must be Lipschitz-continuous, which implies that their variances must be bounded, e.g., there exists $\sigma_{\nabla f} \in \mathbb{R}$ s.t. $\mathbb{E}_{\boldsymbol{h}}[\|\nabla \widehat{F}(\boldsymbol{x}, \boldsymbol{y}; \boldsymbol{h})\|^2] \leq \|\nabla \widehat{F}(\boldsymbol{x}, \boldsymbol{y}; \boldsymbol{h})\|_\infty^2 = \sigma_{\nabla f}$ where the middle expression is well-defined since $\nabla \widehat{F}$ is Lipschitz-continuous over its compact domain.

With all of this machinery in place, we can now extend nested SGDA to compute recursive Stackelberg equilibria in zero-sum Markov Stackelberg games (Algorithm 2; Appendix C). In the usual case, when the policy parameterization does not represent the space of *all* policies $\mathcal{A}^{\mathcal{S}} \times \mathcal{B}^{\mathcal{S}}$, this result should be understood as convergence to the recursive Stackelberg equilibria of a game in which the players' action spaces are restricted to the parameterized policies.

**Theorem 4.1.** *Let $\mathcal{M}$ be a convex-concave zero-sum Markov Stackelberg game. Under Assumption 4, for any $\varepsilon, \delta \geq 0$, if nested policy gradient descent ascent (Algorithm 2, Appendix C) is run with inputs that satisfy for all $t \in \mathbb{N}_+$, $\eta_{\boldsymbol{x}}^{(t)}, \eta_{\boldsymbol{x}}^{(t)} \in \Theta(1/\sqrt{t+1})$, and outputs $(\boldsymbol{x}^*, \boldsymbol{y}^*, \boldsymbol{\lambda}^*)$, then in expectation over all runs of the algorithm (i.e., sample paths of $\boldsymbol{\theta}$ and $\boldsymbol{\phi}$), the policy profile $(\boldsymbol{\pi_{x^*}}, \boldsymbol{\pi_{y^*}})$ is an $(\varepsilon + \delta, \delta)-$recursive SE after $\tilde{O}(1/\varepsilon^2\delta^2)$ oracle calls.*

# 5 Application: Reach-Avoid Problems

In this section, we endeavor to apply our algorithms to a real-world application, namely reach-avoid problems. In a reach-avoid problem (e.g., [29, 32]), an agent seeks to reach one of a set of targets—achieve *liveness*—while avoiding obstacles along the way—ensuring *safety*. Reach-avoid problems have myriad applications, including network consensus problems [42], motion planning [21, 41], pursuit-evasion games [30, 44], autonomous driving [43], and path planning [80], to name a few.

The obstacles in a reach-avoid problem are not necessary stationary; they may move, either randomly or deliberately, around the environment. When the obstacles' movement is random, the problem can be modeled as an MDP. But when their movement is deliberate, so that they are more like a rational opponent than a stochastic process, the problem is naturally modeled as a zero-sum game, where the agent—the protagonist—aims to reach its target, while an antagonist—representing the obstacles—seeks to prevent the protagonist from doing so. Past work has modeled these games as simultaneous-move (e.g., [29], [32]), imposing what should be a hard constraint—that the agent cannot collide with any of the obstacles—as a soft constraint in the form of large negative rewards.

Using the framework of zero-sum Markov Stackelberg games, we model this hard constraint properly, with the leader as the antagonist, whose movements impose constraints on the moves of the follower, the protagonist. We then use nested policy GDA to compute Stackelberg equilibria and simultaneous SGDA to compute GNE, and show experimentally that the protagonist learns stronger policies in the sequential (i.e., Stackelberg) game than in the simultaneous.

A (discrete-time discounted infinite-horizon continuous state and action) *reach-avoid game* $(l, \mathcal{S}, \mathcal{T}, \mathcal{V}, \mathcal{A}, \mathcal{B}, \mu, r, \boldsymbol{h})$ comprises two players, the *antagonist* (or $\boldsymbol{a}$-player) and the *protagonist* (or $\boldsymbol{b}$-player), each of whom occupies a state $\boldsymbol{s_a}, \boldsymbol{s_b} \in \mathcal{S}$ in a state space $\mathcal{S} \subset \mathbb{R}^l$, for some $l \in \mathbb{N}$. The protagonist's goal is to find a path through the safe set $\mathcal{V} \subset \mathcal{S} \times \mathcal{S}$ that reaches a state in the target set $\mathcal{T} \subset \mathcal{V}$, while steering clear of the avoid set $\overline{\mathcal{V}} = \mathcal{S} \times \mathcal{S} \setminus \mathcal{V}$. This safe and avoid set formulation is intended to represent capture constraints, which have been the focus of the reach-avoid literature [80].

Initially, the players occupy some state $\boldsymbol{s}^{(0)} \sim \mu \in \Delta(\mathcal{V})$ drawn from an initial joint distribution $\mu$ over all states, excluding the target and avoid sets. At each subsequent time-step $t \in \mathbb{N}_+$, the antagonist (resp. protagonist) chooses $\boldsymbol{a}^{(t)} \in \mathcal{A}$ (resp. $\boldsymbol{b}^{(t)} \in \mathcal{B}$) from a set of possible directions $\mathcal{A} \subseteq \mathbb{R}^l$ (resp. $\mathcal{B} \subseteq \mathbb{R}^l$) in which to move. After both the antagonist and the protagonist move, they receive respective rewards $-r(\boldsymbol{s}^{(t)}, \boldsymbol{a}^{(t)}, \boldsymbol{b}^{(t)})$ and $r(\boldsymbol{s}^{(t)}, \boldsymbol{a}^{(t)}, \boldsymbol{b}^{(t)})$. Then, either the game ends, with probability $1 - \gamma$, for some discount rate $\gamma \in (0, 1)$, or the players move to a new state $\boldsymbol{s}^{(t+1)} \doteq \boldsymbol{h}(\boldsymbol{s}^{(t)}, \boldsymbol{a}, \boldsymbol{b}) = \left( \boldsymbol{h_a}(\boldsymbol{s_a}^{(t)}, \boldsymbol{a}), \boldsymbol{h_b}(\boldsymbol{s_b}^{(t)}, \boldsymbol{b}) \right)$, as determined by their respective displacement functions $\boldsymbol{h_a} : \mathcal{S} \times \mathcal{A} \to \mathcal{S}$ and $\boldsymbol{h_b} : \mathcal{S} \times \mathcal{B} \to \mathcal{S}$. We can express this deterministic transition as the following probability transition function $p(\boldsymbol{s}' \mid \boldsymbol{s}, \boldsymbol{a}, \boldsymbol{b}) \doteq \mathbb{1}_{\boldsymbol{s}'}(\boldsymbol{h}(\boldsymbol{s}, \boldsymbol{a}, \boldsymbol{b}))$.

We define the feasible action correspondence $\mathcal{C}(\boldsymbol{s}, \boldsymbol{a}) \doteq \{\boldsymbol{b} \in \mathcal{B} \mid \boldsymbol{\alpha}(\boldsymbol{s}, \boldsymbol{a}, \boldsymbol{b}) \geq \boldsymbol{0}\}$ via a vector-valued *safety constraint function* $\boldsymbol{\alpha} : \mathcal{S}^2 \times \mathcal{S} \times \mathcal{S} \to \mathbb{R}^d$, which outputs a subset of the protagonist's actions in the safe set, i.e., for all $(\boldsymbol{s}, \boldsymbol{a}) \in \mathcal{S}^2 \times \mathcal{A}, \mathcal{C}(\boldsymbol{s}, \boldsymbol{a}) \subseteq \{\boldsymbol{b} \in \mathcal{B} \mid \boldsymbol{h}(\boldsymbol{s}, \boldsymbol{a}, \boldsymbol{b}) \in \mathcal{V}\}$. Note that we do not require this inclusion to hold with equality; in this way, the protagonist can choose to restrict itself to actions far from the boundaries of the avoid set, thereby increasing its safety, albeit perhaps at the cost of liveness. Overloading notation, we define the feasible policy correspondence $\mathcal{C}(\boldsymbol{\pi_a}) \doteq \{\boldsymbol{\pi_b} : \mathcal{S} \to \mathcal{B} \mid \boldsymbol{\pi_b}(\boldsymbol{s}) \in \mathcal{C}(\boldsymbol{s}, \boldsymbol{\pi_a}(\boldsymbol{s})), \text{for all } \boldsymbol{s} \in \mathcal{S}\}$.

We consider two forms of reward functions. The first, called the *reach probability reward*, $r(\boldsymbol{s}, \boldsymbol{a}, \boldsymbol{b}) = \mathbb{1}_{\mathcal{T}}(\boldsymbol{s_b})$, is an indicator function that awards the protagonist with a payoff of $1$ if it enters the target set, and $0$ otherwise. Under this reward function, the cumulative payoff function (i.e., the expected value of these rewards in the long term) represents the probability that the protagonist reaches the target, hence its name. The second reward function is the *reach distance reward function*, $r(\boldsymbol{s}, \boldsymbol{a}, \boldsymbol{b}) = -\min_{\boldsymbol{s}' \in \mathcal{T}} \|\boldsymbol{s_b} - \boldsymbol{s}'\|^2$, which penalizes the protagonist based on how far away it is from the target set. With all these definitions in hand, we can now cast the reach-avoid game as a zero-sum Markov Stackelberg game $(2l, l, l, d, \mathcal{S}, \mathcal{A}, \mathcal{B}, \mu, r, \boldsymbol{\alpha}, p, \gamma)$.

The next assumption ensures that 1. under the reach probability reward function, a reach-avoid game is a convex-*non*-concave zero-sum Markov Stackelberg game (i.e., the marginal function $\boldsymbol{x} \mapsto u^*(\boldsymbol{x})$ is convex, and the cumulative payoff function $\boldsymbol{y} \mapsto u(\boldsymbol{x}, \boldsymbol{y})$ is *non*-concave, for all $\boldsymbol{x} \in \mathcal{X}$); and 2. under the reach distance reward function, a reach-avoid game is a convex-concave zero-sum Markov Stackelberg game. Furthermore, a Markov perfect GNE is guaranteed to exist under this assumption, assuming the reach distance reward but not under the reach probability distance.[10]

To state this assumption, for convenience, we model the leader's policy $\boldsymbol{\pi_a}(\boldsymbol{s}) \doteq \boldsymbol{x} \boldsymbol{s_a}$ as parameterized by $\boldsymbol{x} \in \mathcal{X} \subset \mathbb{R}^{l \times l}$, and the follower's policy $\boldsymbol{\pi_b}(\boldsymbol{s}) \doteq \boldsymbol{y} \boldsymbol{s_b}$ as parameterized by $\boldsymbol{y} \in \mathcal{Y} \subset \mathbb{R}^{l \times l}$. Note also that we assume decentralized, play, meaning the players learn only from their own state and rewards, and maintain their policies independently of one another.

**Assumption 5** (Convex-Concave Reach-Avoid Game). *1. The state space $\mathcal{S}$ and the target set $\mathcal{T}$ are non-empty and convex; 2. the action spaces $\mathcal{A}, \mathcal{B}$ are non-empty, compact and convex; 3. the displacement functions $\boldsymbol{h_a}, \boldsymbol{h_b}$ are affine; 4. $\boldsymbol{a} \mapsto \boldsymbol{\alpha}(\boldsymbol{s}, \boldsymbol{a}, \boldsymbol{b})$ is log-convex for all $\boldsymbol{b} \in \mathcal{B}$, and $\boldsymbol{b} \mapsto \boldsymbol{\alpha}(\boldsymbol{s}, \boldsymbol{a}, \boldsymbol{b})$ for all $(\boldsymbol{s}, \boldsymbol{a}) \in \mathcal{S} \times \mathcal{A}$; 5. the players' parameter spaces $\mathcal{X}$ and $\mathcal{Y}$ are non-empty, compact, and convex; and 6. the players policies are biaffine, i.e., $\boldsymbol{\pi_x}(\boldsymbol{s}) \doteq \boldsymbol{x} \boldsymbol{s_a}$ and $\boldsymbol{\pi_y}(\boldsymbol{s}) \doteq \boldsymbol{x} \boldsymbol{s_b}$.*

Part 1 is a standard assumption commonly imposed on reach-avoid games (see, for instance Fisac et al. [29]). Part 3 is satisfied by natural displacement functions of the type $\boldsymbol{h}(\boldsymbol{s}, \boldsymbol{a}, \boldsymbol{b}) = \boldsymbol{s} + \beta(\boldsymbol{a}, \boldsymbol{b})$, for some $\beta \in \mathbb{R}$, which is a natural characterization of all displacement functions with constant velocity $\beta$, when $\mathcal{A} = \mathcal{B} \subseteq \{\boldsymbol{z} \in \mathcal{S} \mid \|\boldsymbol{z}\| = 1\}$. Part 4 is satisfied by various action correspondences, such as $\boldsymbol{\alpha}(\boldsymbol{s}, \boldsymbol{a}, \boldsymbol{b}) \doteq \exp\{\min_{\boldsymbol{s}' \in \overline{\mathcal{V}}} \|(\boldsymbol{h_a}(\boldsymbol{s_a}, \boldsymbol{a}), \boldsymbol{s_b}) - \boldsymbol{s}'\|\} - 1 - \|\boldsymbol{h_b}(\boldsymbol{s_b}, \boldsymbol{b}) - \boldsymbol{s_b}\|$, which shrinks

---

[10]The existence of Markov perfect GNE, and hence GNE, is guaranteed by a straightforward generalization of Shapley's [65] result on the existence of Markov perfect Nash equilibria in zero-sum Markov games.

the space of actions exponentially as the protagonist approaches the antagonist, and can thus be interpreted as describing a safety-conscious protagonist. The following theorem states the convex-concavity properties of reach-avoid games, and shows polynomial-time computability of recursive SE under Assumption 5. Note that for the reach probability reward function, it is not possible to obtain a polynomial-time convergence result, result since the rewards are not even continuous.

**Theorem 5.1.** *Under the reach distance (resp. reach probability) reward function, any reach-avoid game for which Assumption 5 hold is convex-concave (resp. convex-non-concave). Moreover, if $\alpha$ is Lipschitz-continuous, then nested SGDA is guaranteed to converge in such games to recursive SE policies in polynomial time.*

**Experiments.** We ran a series of experiments on reach-avoid problems,[11] which were designed to assess the efficacy of policies learned in a Stackelberg game formulation as compared to those learned in a simultaneous-move game formulation, assuming complex, i.e., neural, policy parameterizations.

We consider a variant of the two-player differential game introduced in Isaacs [38], played by two Dubins cars. A Dubins car is a simplified model of a vehicle with a constant forward speed $\nu$ and a constrained turning radius $\omega$. We model both the protagonist and antagonist as Dubins cars [38] moving around a 2-dimensional state space. The target set is a select subset of the state space, while the avoid set, which defines the safe set, is a ball around the antagonist.

We experiment with only the reach distance, not the reach probability, reward function. In all safe states, the reward is actually a penalty, measuring the protagonist's distance to the target set, while a bonus $\beta$ is awarded upon reaching a target, at which point the game ends. This reward function suffices for our Stackelberg game setup, which enforces the hard constraint that the protagonist cannot move into the avoid set. In our simultaneous-move game setup, we achieve a similar effect by enhancing the aforementioned reward function with a large penalty $(-\beta)$ whenever the protagonist touches the avoid set. As in the case of reaching the target, touching the avoid set ends the game.

We note that this reach-avoid game is not actually a continuous game, as there is a discontinuity in the reward function when the target is reached. Additionally, it is possible for the antagonist to be "cornered," meaning left with an empty set of feasible actions (in which case the game ends). For these reasons, recursive SE are not guaranteed to exist in our setup.

Our experiments were run on a 7x7 square grid, with the target set $\mathcal{T}$ a closed ball of radius 1 centered along the lower edge, and the avoid set $\overline{\mathcal{V}}$ a closed ball of radius 0.3 around the antagonist. We set the bonus (resp. penalty) for reaching the target (resp. avoid set) $\beta = 200$, $\omega = 30°$, and $\nu = 0.25$.

Using this experimental setup, we train two agents by playing two games, the Stackelberg and simultaneous-move variants of the reach-avoid game, using nested policy GDA and SGDA, respectively. We evaluate the protagonists' policies to assess their safety and liveness characteristics.

To assess liveness, we ran our agents against an opponent that plays actions sampled uniformly at random. To assess safety, we ran our agents against an

| Match-up | Outcome | Mean win length | Loss/draw length |
|---|---|---|---|
| GNE vs. random | 47 W, 18 L, 35 D | $23.23 \pm 7.53$ | $33.71 \pm 19.31$ |
| SE vs. random | 95 W, 2 L, 3 D | $18.16 \pm 3.69$ | $33.0 \pm 20.8$ |
| GNE vs. chaser | 0 W, 100 L, 0 D | N/A | $8.53 \pm 1.90$ |
| SE vs. chaser | 63 W, 36 L, 1 D | $21.63 \pm 5.04$ | $11.06 \pm 7.71$ |

Table 1: Game results summary for GNE and SE agents.

opponent who chases them, always taking actions that minimize their distance. Table 1 reports the number of wins (W), losses (L), and draws (D), and average game lengths, of 100 games against each opponent. An agent, playing the role of the protagonist, wins when it reaches the target set. A GNE agent loses if it enters $\overline{\mathcal{V}}$, while a Stackelberg agent loses if it finds itself cornered. The game is a draw if neither player wins or loses within 50 time steps.

We find that the SE agent outperforms the GNE agent by a large margin. The SE agent wins almost all of its games against random, and roughly ⅔ of its games against the chaser, while the GNE agent wins only half of its games against random, and none of its games against the chaser. Moreover, even when the SE agent loses or draws, it tends to stay alive longer than the GNE agent. Not only does our Stackelberg approach outperform GNE, it is tractable as well. Our methods thus seem to offer a promising path to further progress solving the myriad of robotic applications of reach-avoid.

---

[11]Our code is found at: https://github.com/arjun-prakash/stackelberg-reach-avoid.

# 6 Acknowledgments

Denizalp Goktas was supported by a JP Morgan AI fellowship. Arjun Prakash was partially supported by ONR N00014-22-1-2592 and the Quad Fellowship.

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
