# OpenReview forum: "Convex-Concave Zero-Sum Markov Stackelberg Games"
_NeurIPS.cc/2023/Conference — NeurIPS 2023 poster_

### Official Review · Reviewer_5fZF · 2023-06-19

**Soundness:** 3 good
**Presentation:** 3 good
**Contribution:** 3 good
**Rating:** 6
**Confidence:** 4

**Summary:**

This paper develops policy gradient methods using stochastic gradient estimates from the trajectories of play for computing in polynomial time Stackelberg equilibria in convex-concave games, most notably including a certain class of reach-avoid problems. The authors also demonstrate through experiments the benefit of Stackelberg equilibria over their simultaneous-move counterparts in reach-avoid problems.

**Strengths:**

Prior work by Goktas and Greenwald [23] addressed the problem of computing Stackelberg equilibria in a certain class of min-max optimization problems using an exact gradient oracle. The present paper extends those prior results by providing polynomial-time guarantees even when only stochastic gradient estimates from the trajectories of play are available, filling a gap that was left in prior work. Moreover, the paper nicely motivates the underlying problem, providing a number of concrete applications, especially with the experiments of Section 5 that demonstrate the benefit of computing Stackelberg versus Nash equilibria in a certain class of problems. Overall, the paper addresses an important problem with a theoretically sound approach, and provides non-trivial improvements over the prior state of the art.

The paper is also well-written, and the ideas and concepts are clearly exposed. The results are also accurately placed into the existing literature.


**Weaknesses:**

The main weakness of this paper is that the overall contribution is arguably somewhat incremental in light of prior work, especially papers [23] and [25]. The techniques employed to extend the analysis from exact to stochastic gradient information are overall standard, and the main results are certainly not surprising. That being said, the paper closes a theoretical gap left from prior work, and I do believe that the technical contribution is non-trivial.


**Questions:**

Some minor issues:
1. Line 12: Missing space before the first set of references
2. Line 50: A Stackelberg equilibria -> A Stackelberg equilibrium
3. Lines 59-60: Technically this is a pseudo-polynomial time algorithm since the dependence on $1/\epsilon$ is polynomial
4. Why are you using the notation $\mathbb{N}_+$ instead of just $\mathbb{N}$? (those are presumably the same sets)
5. Lines 132 - 134: The game being convex-concave is surely not the only condition that makes the problem amenable to first-order methods
6. Theorem 3.1: then, in expectation over all runs of the algorithm, then (I would remove the second repetition of the word "then")
7. Line 673: Missing dot
8. Line 694: as as
9. There are many overfull equations in pages 21-23 of supplementary; I strongly encourage to fix those
10. Overall there are missing punctuation marks in the equations throughout the entire paper
11. Line 839: the argument is incomplete
12. It would be helpful to introduce the notion of stochastic convexity since it is quite non-standard


**Limitations:**

The authors have adequately addressed all limitations.

---

> ### Author Rebuttal · Authors · 2023-08-10
>
> Thank you for your review, and for taking the time to point out several minor issues!
>
> **Regarding the weaknesses.**
>
> Although our results are an extension of the convergence guarantees provided by Goktas et al. [1] to a setting with a stochastic first-order oracle, we view our main contribution to be the convergence of nested policy gradient methods in zero-sum stochastic Stackelberg games, and the applicability of these methods to reach-avoid problems. Our results in this vein make use of novel assumptions (based on the extensively studied notion of convex stochastic dominance) and proof techniques, to prove that under suitable assumptions zero-sum stochastic Stackelberg games can be shown to be convex-concave w.r.t. to the parameters of the leader and follower’s policy parameters, thereby allowing us to obtain convergence to a recursive Stackelberg equilibrium. Perhaps more importantly, in Appendix C, we show that the important class of reach avoid problems [2] can be modeled as convex-concave zero-sum stochastic Stackelberg games (see Theorem C.1. in Appendix C). Using this characterization, we obtain novel polynomial-time solution methods for these problems, which as we show in experiments, outperform known solutions/methods (i.e., Nash equilibrium).
>
> **References**
>
> [1] Goktas, Denizalp, and Amy Greenwald. "Convex-concave min-max stackelberg games." Advances in Neural Information Processing Systems 34 (2021): 2991-3003.
>
> [2] Jaime F. Fisac, Mo Chen, Claire J. Tomlin, and S. Shankar Sastry. Reach-avoid problems with time-varying dynamics, targets and constraints. In Proceedings of the 18th International
> Conference on Hybrid Systems: Computation and Control, HSCC ’15, page 11–20, New York, NY, USA, 2015. Association for Computing Machinery

---

> > ### Comment · Reviewer_5fZF · 2023-08-10
> >
> > I thank the authors for the detailed reply.

---

### Official Review · Reviewer_sSzA · 2023-07-02

**Soundness:** 2 fair
**Presentation:** 2 fair
**Contribution:** 2 fair
**Rating:** 5
**Confidence:** 4

**Summary:**

This paper considers a convex-concave  min-max Stackelberg game and proposes algorithm which converges to the Stackelberg equilibrium. The paper also proposes a policy-gradient based mechanism which also converges to the Stackelberg equilibrium for the Markov game.

**Strengths:**

Stackelberg game is an important problem in multi-agent setup and also for hyper-parameter tuning. Thus, this paper seeks to address some interesting questions and close the gap. The results are clean and thus have value.

**Weaknesses:**

1. The contribution part is not clear and it seems that they are minor. There are recent works on First-order methods for Stackelberg-game [A1]. However, the paper has not compared with those approaches.

[A1]. Maheshwari, Chinmay, S. Shankar Sasty, Lillian Ratliff, and Eric Mazumdar. "Convergent first-order methods for bi-level optimization and stackelberg games." arXiv preprint arXiv:2302.01421 (2023).

2. The paper has made a lot of Assumptions in order to achieve results (specially for MDP). However, can they be satisfied in practice? Specially, it seems that the action correspondence  is concave in $x$ is very strong assumption. Why $f(x,y)$ is convex in $y$ means that it must be affine?

3. For general simultaneous game, average gradient descent average converges to the saddle point for min-max game. This result also shows that average indeed converges to max-min solution. Discussion with the simultaneous game and the technical difficulties must be included. Further, the algorithm seems to avoid simultaneous game complications by finding the solution or the saddle point $(y, \lambda)$ for the lower-level game (assuming a saddle-point solver or simultaneous game setup whose results are kind of well-known). Hence, technical contributions seem to be limited. Further, there is no last-iterate convergence guarantee where the min-max game (simultaneous play) already have achieved that (using extra gradient, or optimistic gradient)

4. All the algorithms seem to have large sample complexity, given the fact that it relies on finding the lower-level solution for each upper-level point $x$. How this algorithm can be implemented in practice? Will leader just pause in updating its strategy while followers updating their strategies?

4. The paper is not very well-written. For example, the MDP setup is not at all clear. Does the leader and follower take decisions in turn at the same state? Does the state transition to the next state after both the leader and follower take actions?

5. The numerical setup should also be expanded.

**Questions:**

1. The paper uses policy-gradient approach. However, it relies on the generator model where a state-action pair is drawn from the state-action occupancy measure? Is it practical? Further, the paper seems to work on continuous state-space. The convergence result is for finite state-space only, while limited results exist for function approximation setup (severely relies on the realizability assumption). However, this paper does not consider such. Does the policy-gradient algorithm have any convergence guarantee? Is the action-space also continuous? Standard policy-gradient does not work with the continuous action.

2. Two-time scale approximations are usually employed for Stackelberg game to reduce the sample complexity. Can it be done in this case as well. For example, see the recent work [A2, A3]

[A2]. Li, Haochuan, Farzan Farnia, Subhro Das, and Ali Jadbabaie. "On convergence of gradient descent ascent: A tight local analysis." In International Conference on Machine Learning, pp. 12717-12740. PMLR, 2022.

[A3]. Lin, Tianyi, Chi Jin, and Michael Jordan. "On gradient descent ascent for nonconvex-concave minimax problems." In International Conference on Machine Learning, pp. 6083-6093. PMLR, 2020.

3. Are Theorems 3.1 and 4.1 correct? In the averaging of $y$, why the numerators are multiplied by $\eta_x$ instead of $\eta_y$? I tried to find it in the proof, however, I could not. Can the authors point where it has been shown that the average is done over $y_t$?

**Limitations:**

Not as such.

---

> ### Author Rebuttal · Authors · 2023-08-10
>
> Thank you for your review!
>
> 1) Our model is not captured or covered by [A1], because in [A1], the action space of the followers does not depend on the leader’s action. Due to space limitations, we were unable to include a discussion of bi-level optimization more broadly; however, we can use the additional page in the camera-ready version to provide further context.
>
> 2) The assumptions that we present are specifically satisfied by reach-avoid problems (see Appendix C and Theorem C.1.). Reach-avoid problems are important because of their applicability to robotics problems, such as autonomous driving. The fact that our assumptions capture this important class of problems substantiates their utility. Further, these assumptions enable polynomial-time computational guarantees, and thus open the door to roboticists to build practical solutions for this important class of problems.
>
> 3) $f$ is assumed to be convex in (x,y) and concave in $y$; as such $f$ must also be affine in $y$.
>
> 4) Although we agree that proving convergence in last iterates would be of interest, even in simultaneous settings, assuming access only to a stochastic first-order oracle, convergence in last iterates is *not* known. The results for extragradient descent ascent hold only for *exact* first-order oracle settings [1].
>
> 5) Additionally, although our proof techniques are an extension of the convergence technique provided by Goktas et al. [2] to a setting with a stochastic first-order oracle, we want to note that our main technical novelty pertains to the convergence of nested policy gradient methods in zero-sum stochastic Stackelberg games, and their application to reach-avoid problems.
> The algorithm by nature is sequential, meaning that the leader has to wait for the follower to best respond, so as to converge to a (recursive) Stackelberg equilibrium. Our experiments suggest that our algorithm is sample efficient in practice.
>
> 6) The leader and follower take action sequentially. The leader first commits to an action, after which the follower, having observed the leader’s action, takes its own action. Once both players have made their moves, the game moves to a new state. This description can be found in lines 29-33. For more explanation of such dynamics, we refer you to the introduction of Goktas et al. [2].
>
> **Regarding your questions.**
>
> 1. Our generator model is not based on the state-action visitation distribution, but rather the history distribution. This model is realistic, since simulating a trajectory of the game is possible given the policies of both players. Additionally, we note that our convergence results apply to both continuous and discrete state spaces, but not to continuous action spaces.
>
> 2. Two-time scale approaches do not allow for convergence to a Stackelberg equilibrium, when the leader’s action determines the action space of the follower (See Example 3.3. in [4]).
>
> 3. This is a typo. The $\eta$ in the numerators should match the variables. Note that the average-iterate convergence result for $\mathbf{y}^{(t)}$ is due to Theorem 3.15 of Nemirosvki [3] as described in lines 768-770 of Appendix E. We realize this point may not be clear to the reader, so we will add an explicit reference in the camera-ready version.
>
> **References**
>
> [1] Golowich, Noah, et al. "Last iterate is slower than averaged iterate in smooth convex-concave saddle point problems." Conference on Learning Theory. PMLR, 2020.
>
> [2] Goktas, Denizalp, and Amy Greenwald. "Convex-concave min-max Stackelberg games." Advances in Neural Information Processing Systems 34 (2021): 2991-3003.
>
> [3] Nemirovski, Arkadi, et al. "Robust stochastic approximation approach to stochastic programming." SIAM Journal on optimization 19.4 (2009): 1574-1609.
>
> [4] Goktas, Denizalp, and Amy Greenwald. "Gradient Descent Ascent in Min-Max Stackelberg Games." arXiv preprint arXiv:2208.09690 (2022).

---

> > ### Comment · Reviewer_sSzA · 2023-08-11
> > **Follow up Questions**
> >
> > Thank you for your detailed answers. I now understand your contributions. I have a few more follow-up questions.
> >
> > 1. The authors repeatedly mentioned the convergence of the nested policy gradient algorithm to the Stackelberg equilibrium as their main contribution. However, I have a few comments on this.
> >
> >     a. In order to prove the convergence the authors rely on Theorem 3.1. However, that heavily depends on the structure of the function such as convex in $x$, and affine in $y$. I am wondering whether this restricts the practicality of this approach. For example, it is well known that value function in general is not concave in the policy space. The authors did mention sufficient conditions on reward, I am wondering for which settings those would hold.
> >
> >    b. This leads to the second question. The proof of Lemma 1 is not correct. The authors have used the concavity of $v$ to prove the concavity of $q$, and then the concavity of $q$ to prove the concavity of $v$.
> >
> > 2. The reviewer is also confused with the structure of the constraint for the MDP. Is it related to the CMDP setup [1,2] (meaning the at some cumulative cost/utility has to be less than or equal to some threshold) or is it something else, like at every state a constraint is required to be satisfied? The paper has considered $\underline{g}$ which means taking minimum across all the states. How can one evaluate this value as one needs to evaluate the policy at every possible state?
> >
> > 3. I do not understand the statement ``Our generator model is not based on the state-action visitation distribution, but rather the history distribution". The algorithm (Algorithm 2) needs to generate samples for the current policy $\pi_x,\pi_y$. Hence, it needs to access any type of generator or simulator model (whether it depends on the history distribution is immaterial). Furthermore, I don't think the gradient computed in Algorithm 1 is an unbiased estimator as claimed by the authors (if it is an infinite-dimensional). Please see [3] on how to ensure unbiasedness in the Q-function evaluation.
> >
> > [1]. Efroni, Y., Mannor, S. and Pirotta, M., 2020. Exploration-exploitation in constrained mdps. arXiv preprint arXiv:2003.02189.
> >
> > [2]. Ghosh, A., Zhou, X. and Shroff, N., 2022. Provably efficient model-free constrained rl with linear function approximation. Advances in Neural Information Processing Systems, 35, pp.13303-13315.
> >
> > [3]. Zhang, K., Koppel, A., Zhu, H. and Basar, T., 2020. Global convergence of policy gradient methods to (almost) locally optimal policies. SIAM Journal on Control and Optimization, 58(6), pp.3586-3612.

---

> > > ### Author Response · Authors · 2023-08-13
> > > **Answers to Follow-Up**
> > >
> > > Thank you for your reply! We really appreciate the time you are putting into reviewing our work!
> > >
> > > Regarding the points you have made.
> > >
> > > **1.**
> > > **a.** Theorem 3.1 holds for any convex-concave min-max Stackelberg game for which we only have access to a noisy estimate of the gradient and is thus not necessarily restricted to stochastic Stackelberg games (unlike Theorem 4.1). As such, Theorem 3.1, has applications beyond stochastic Stackelberg games (which we discuss below), and also has applications to a number of problems of interest which are modeled as convex-concave min-max Stackelberg games such as resource allocation and automated test generation (see for instance [1] or [2]).
> > >
> > > We understand your point that the state-value function of an MDP is in general not concave in the parameters of the policy, and it only satisfies a gradient dominance condition. However, this result holds in extremely broad settings (i.e., with no assumption on the transition probability function); in a number of interesting problems, the state-value function can be guaranteed to be concave in the parameters of the policy, as we show is the case with reach-avoid problems [3]. An important property of reach-avoid problems is that the transition function is deterministic and affine in the actions of the players (see Appendix C for more details), which allows us to show that the game is convex-concave.
> > >
> > > Additionally, we would like to note that Assumption 2 is only a sufficient and *not* necessary for the game to be convex-concave. Moreover, our results also generalize directly to what we would call convex-incave min-max Stackelberg games (see Footnote 4 for an explanation). This class of games covers any zero-sum stochastic Stackelberg game in which for the leader, the marginal state-value function (see line 248 for a definition) is convex, and the follower’s problem is a discrete state/action MDP.
> > >
> > > **1.b.** We disagree with you that Lemma 1 is incorrect. First of all, we would like to note that even if you disagree with our proof of the concavity of the state-value function, it is a known result in the economics literature (see for instance Theorem 1 of [4] and the discussion above Theorem 1). We include a proof of this fact in Lemma 1 only for completeness. We chose to present a compact version of the argument since it would otherwise require us to replicate steps in the proof of the Banach fixed point theorem. All of that said, our argument is simply an inductive one, and one simple way to understand it is as follows:
> > >
> > > Consider the policy improvement process which consists of applying the Bellman expected operator iteratively, which is known to converge to the state-value function associated with a given policy. Suppose that we initialize the value function for this process to be a continuous, concave, bounded function (base case). Our proof then shows that the Bellman expected operator preserves concavity (inductive step), hence the process can only converge to a continuous, concave, bounded state-value function. You can find a similar argument in the proof of Theorem 1 of [4]. We hope this helps clarify our results. If you still disagree with us for some reason, we would be interested to understand your objection in more detail, since incorrectness of this result would invalidate a large body of results in mathematical economic theory.
> > >
> > > **2.** The constraints in our setting are different than the way constraints are traditionally represented in the constrained MDP literature. In constrained MDPs, there are cumulative cost functions, which are required to satisfy a certain constraint; in our setting, the constraints are not cumulative, but rather required to be satisfied only locally. That is, in your own terms, “at every state a constraint is required to be satisfied”. Since the constraints are much less computationally complex in our setting, $\underline{g}$ can be easily computed in discrete-state MDPs by just checking the value of the constraint at each state, and in continuous-state settings by simply running gradient descent.
> > >
> > > **3.** The goal of our statement was to point out that the gradient can be computed by unrolling trajectories of play. This is a standard assumption in literature on learning in games (see for instance [6]). It is our understanding that Zhang et al’s [7] goal is to obtain an unbiased gradient estimate using only *finite* horizon trajectories; however, in line with the learning in games literature (see for instance [6]), we do not restrict ourselves to finite horizon trajectories, and in such cases the REINFORCE estimator remains unbiased (see Lemma 2 of [6]). That said, we do agree that Zhang et al’s gradient estimate (Equation 3.6 of [7]) has better properties and we are happy to use that estimate should you think it is more appropriate. Our results hold with either estimate.

---

> > > > ### Author Response · Authors · 2023-08-13
> > > > **References**
> > > >
> > > > [1] Badithela, Apurva, et al. "Synthesizing reactive test environments for autonomous systems: testing reach-avoid specifications with multi-commodity flows." 2023 IEEE International Conference on Robotics and Automation (ICRA). IEEE, 2023.
> > > >
> > > > [2] Goktas, Denizalp, and Amy Greenwald. "Gradient Descent Ascent in Min-Max Stackelberg Games." arXiv preprint arXiv:2208.09690 (2022).
> > > >
> > > > [3] Jaime F. Fisac, Mo Chen, Claire J. Tomlin, and S. Shankar Sastry. Reach-avoid problems with time-varying dynamics, targets and constraints. In Proceedings of the 18th International
> > > > Conference on Hybrid Systems: Computation and Control, HSCC ’15, page 11–20, New York, NY, USA, 2015. Association for Computing Machinery.
> > > >
> > > > [4] Alp E. Atakan. Stochastic convexity in dynamic programming. Economic Theory, 22(2):399
> > > > 447–455, 2003
> > > >
> > > > [5] Denizalp Goktas, Sadie Zhao, and Amy Greenwald. Zero-sum stochastic Stackelberg games. Advances in Neural Information Processing Systems, 35:11658–11672, 2022
> > > >
> > > > [6] Constantinos Daskalakis, Dylan J Foster, and Noah Golowich. Independent policy gradient
> > > > methods for competitive reinforcement learning. Advances in neural information processing systems.
> > > >
> > > > [7] Zhang, Kaiqing, et al. "Global convergence of policy gradient methods to (almost) locally optimal policies." SIAM Journal on Control and Optimization 58.6 (2020): 3586-3612.

---

> > > > > ### Comment · Reviewer_sSzA · 2023-08-15
> > > > > **Thank you**
> > > > >
> > > > > I thank the authors for their detailed replies. I do not have any more questions. I have increased my score to 5. The results are quite interesting and first of a kind to the RL community even though the theoretical contributions are little limited.

---

### Official Review · Reviewer_yZL6 · 2023-07-05

**Soundness:** 3 good
**Presentation:** 3 good
**Contribution:** 3 good
**Rating:** 5
**Confidence:** 4

**Summary:**

The authors propose a policy gradient method to solve the zero-sum stochastic Stackelberg game from noisy gradient estimates computed from observed trajectories of play. When the games are convex-concave, the authors prove that the proposed algorithms converge to Stackelberg equilibrium in polynomial time.

**Strengths:**

The authors have completed a thorough theoretical analysis of the proposed policy gradient method and prove it converges to Stackelberg equilibrium in polynomial time assuming the game is convex-concave.

**Weaknesses:**

1. The authors claim they solved the convex-concave zero-sum Stackelberg problem but also restrict the application to a reach-avoid problem. The authors mention that "We also prove that reach-avoid problems are naturally modeled as convex-concave zero-sum stochastic Stackelberg games." in the abstract but in fact do not really PROVE it in the manuscript.

2. Besides, as the author mentioned, the reach-avoid game is usually formulated as a single-agent problem and there exist some works formulating it as a two-player zero-sum game.  Why do the authors not address standard zero-sum problems like pricing or allocating goods across agents and time as in reference [25]? Experimenting merely on a reach-avoid game is not enough to demonstrate the effectiveness or superiority of the proposed approach compared to existing methods like Nash equilibrium.

3. The authors make several critical assumptions like convex-concave, convergence assumptions, etc. How feasible these assumptions are in reality and how to validate them are missing. The authors should address these limitations and discuss potential extensions to more realistic scenarios.

**Questions:**

See above.

**Limitations:**

See above.

---

> ### Author Rebuttal · Authors · 2023-08-10
>
> Thank you for your review!
>
> **Regarding the weaknesses.**
>
> We did include a proof of reach-avoid games being an instance of convex-concave zero-sum stochastic Stackelberg games in Appendix C (see Theorem C.1. and the ensuing proof). However, it seems like we failed to add a forward reference to this theorem in the main body of the paper; we will correct this oversight in the camera-ready version. And in the meantime, we are happy to answer any questions you might have about this result, or its proof.
>
> The reason why we do not use our algorithm to solve, for example, the zero-sum stochastic Stackelberg resource allocation game introduced in [1] is because that game is not convex-concave, and as such it lies beyond the scope of our theory and paper. We focused specifically on reach-avoid problems, as they *are* naturally convex-concave (Theorem C.1.).
>
> We refer you to part (2) of our response to common reviewer concerns.
>
> We do provide one simple way of validating the convex-concavity of games, which we use in Appendix C to prove the convex-concavity of reach-avoid problems. In particular, it suffices to check that the problem satisfies Assumption 2.
>
> **References**
>
> [1] Denizalp Goktas, Sadie Zhao, and Amy Greenwald. Zero-sum stochastic stackelberg games. Advances in Neural Information Processing Systems, 35:11658–11672, 2022
>
> [2] Tsaknakis, Ioannis, Mingyi Hong, and Shuzhong Zhang. "Minimax problems with coupled linear constraints: computational complexity, duality and solution methods." arXiv preprint arXiv:2110.11210 (2021).
>
> [3] Goktas, Denizalp, and Amy Greenwald. "Convex-concave min-max stackelberg games." Advances in Neural Information Processing Systems 34 (2021): 2991-3003.
>
> [4] Jaime F. Fisac, Mo Chen, Claire J. Tomlin, and S. Shankar Sastry. Reach-avoid problems with time-varying dynamics, targets and constraints. In Proceedings of the 18th International
> Conference on Hybrid Systems: Computation and Control, HSCC ’15, page 11–20, New York, NY, USA, 2015. Association for Computing Machinery
>
> [5] Badithela, Apurva, et al. "Synthesizing reactive test environments for autonomous systems: testing reach-avoid specifications with multi-commodity flows." 2023 IEEE International Conference on Robotics and Automation (ICRA). IEEE, 2023.

---

> > ### Comment · Reviewer_yZL6 · 2023-08-20
> > **Thanks**
> >
> > I appreciate the authors' thorough response.

---

### Official Review · Reviewer_gXac · 2023-07-07

**Soundness:** 4 excellent
**Presentation:** 3 good
**Contribution:** 3 good
**Rating:** 6
**Confidence:** 4

**Summary:**

The paper considers the setting of convex-concave zero-sum stochastic Stackelberg games. In these games, there are two players, a leader and a follower. The leader's strategies constrain the feasible strategy set of the follower. First, the leader commits to a certain strategy, and then, the follower best-responds to that strategy using a strategy that is feasible.

Previous work mainly focuses on the static version of that game where the utility of the two players is monotone (convex-concave games). The authors define what is a convex-concave stochastic zero-sum stochastic Stackelberg game and then use some machinery from min-max optimization to solve the problem of computing a Stackelberg equilibrium.

---

I acknowledge that I have read and evaluated the rebuttal. The rebuttal answered my concerns and reinforced with my initial positive assessment.

**Strengths:**

* The exposition of previous work and motivation is clear.
* The paper uses standard methods and indicates a good understanding of the min-max optimization and constrained optimization literature.
* The guarantees are for the stochastic setting (i.e., no full information gradient is needed.)

**Weaknesses:**

* Assumption on convexity concavity seems restrictive. Is there any other way to ensure the existence of Stackelberg equilibria AND convergence?
* The results seem a little straightforward given the existing machinery in the literature of min-max optimization.

**Questions:**

* Is the assumption of convexity-concavity necessary to prove convergence? Could some other assumption guarantee it?
* Do you believe that the convergence rates are optimal? Could you use some other optimization method to get better rates?

**Limitations:**

* The main limitation in my opinion is the fact that convexity-concavity is assumed. I think the authors could elaborate more on the justification of such an assumption. (It could also be the case that this assumption is a necessity though.)

---

> ### Author Rebuttal · Authors · 2023-08-10
>
> Thank you for your review!
>
> **Regarding the weaknesses.**
>
> Beyond convex-concave domains, convergence to a Stackelberg equilibrium unfortunately becomes NP-hard [1]. As global convergence guarantees to (recursive) Stackelberg equilibrium have not been obtained for more general zero-sum (stochastic) Stackelberg games under the assumption of a stochastic first-order oracle, we chose to restrict our attention to convex-concave domains.
>
> We want to push back slightly on your comment that our results are straightforward. Although extensions of the convergence guarantees provided by Goktas et al. [2] to a setting with a stochastic first-order oracle may be straightforward, we want to note that our main contribution is regarding the convergence of nested policy gradient methods in zero-sum stochastic Stackelberg games, and their application to reach-avoid problems. Our results in this direction, make use of novel assumptions (based on the extensively studied notion of convex stochastic dominance) and proof techniques, to develop suitable assumptions under which zero-sum stochastic Stackelberg games can be shown to be convex-concave w.r.t. to the parameters of the leader and follower’s policy parameters, allowing us to obtain convergence to a recursive Stackelberg equilibrium. Our assumptions, proof techniques, and results open the door to proving the convergence of policy gradient methods in a new class of continuous state-discrete action space MDPs/Markov games.
>
> Perhaps more importantly, in Appendix C, we show that the well-established class of reach avoid problems [3] can be modeled as convex-concave zero-sum stochastic Stackelberg games (see Theorem C.1. in Appendix C). Using this characterization of reach-avoid problems, we are able to obtain novel polynomial-time solution methods for these problems, which, as we show in experiments, outperform known solutions/methods (i.e., Nash equilibrium).
>
> **Regarding your questions.**
>
> We refer you to part (2) of our response to common reviewer concerns.
>
> We believe that our convergence rates are “nearly” optimal, in the sense that, excluding the use of Nesterov’ momentum or other acceleration methods, our method is optimal. Our convergence rate could be improved by an order of magnitude by using acceleration, but to keep our paper accessible, e.g., to roboticists, and potentially be more impactful, we chose to not present such an algorithm. Additionally, we want to note that the nested nature of our algorithm is essential, because single loop algorithms cannot converge to Stackelberg equilibria in convex-concave zero-sum Stackelberg games with coupled/dependent action spaces (See Example 3.3 in [5]).
>
>
> **References**
>
> [1] Tsaknakis, Ioannis, Mingyi Hong, and Shuzhong Zhang. "Minimax problems with coupled linear constraints: computational complexity, duality and solution methods." arXiv preprint arXiv:2110.11210 (2021).
>
> [2] Goktas, Denizalp, and Amy Greenwald. "Convex-concave min-max stackelberg games." Advances in Neural Information Processing Systems 34 (2021): 2991-3003.
>
> [3] Jaime F. Fisac, Mo Chen, Claire J. Tomlin, and S. Shankar Sastry. Reach-avoid problems with time-varying dynamics, targets and constraints. In Proceedings of the 18th International
> Conference on Hybrid Systems: Computation and Control, HSCC ’15, page 11–20, New York, NY, USA, 2015. Association for Computing Machinery
>
> [4] Badithela, Apurva, et al. "Synthesizing reactive test environments for autonomous systems: testing reach-avoid specifications with multi-commodity flows." 2023 IEEE International Conference on Robotics and Automation (ICRA). IEEE, 2023.
>
> [5] Goktas, Denizalp, and Amy Greenwald. "Gradient Descent Ascent in Min-Max Stackelberg Games." arXiv preprint arXiv:2208.09690 (2022).

---

### Author Rebuttal · Authors · 2023-08-10

We would like to thank all the reviewers for their time!

**Summary of our contributions**: We present polynomial-time first-order methods to compute Stackelberg equilibrium in convex-concave min-max Stackelberg games, assuming access to only a first-order gradient oracle (Theorem 3.1). We then introduce the class of convex-concave zero-sum stochastic Stackelberg games, provide sufficient conditions to validate convex-concavity (Lemma 1 and Lemma 2), and obtain polynomial-time convergence guarantees to recursive competitive equilibrium via a policy-gradient-type algorithm in discrete/continuous state and discrete action convex-concave zero-sum stochastic Stackelberg games (Theorem 4.1). Finally, we show that reach-avoid games [1], which have found important applications in robotics, can be modeled as convex-concave zero-sum stochastic Stackelberg games (Appendix C, Theorem C.1, part 1). Using this result, we obtain polynomial-time solution methods for such games (Appendix C, Theorem C.1, part 2.), and run experiments using neural policy classes.

**Summary of common concerns of reviewers**:
1) Reviewers gXac, sSzA, and 5fZF suggested our contributions could be seen as incremental.
2) Reviewers yZL6 and gXac requested additional motivation for the assumption of convex-concavity.

**Answer to common concerns**:
1) While our results and proof techniques necessitate extensions of Goktas et al.’s [2] results to a setting with a stochastic first-order oracle, we do not consider these extensions to be the main contribution of our work. Rather, our main contribution is the convergence of nested policy gradient methods in a novel class of zero-sum stochastic Stackelberg games, and their application to reach-avoid problems.
Our main results make use of novel assumptions (based on the extensively studied notion of convex stochastic dominance) and proof techniques, to prove that under suitable assumptions, zero-sum stochastic Stackelberg games can be shown to be convex-concave w.r.t. to the parameters of the leader and follower’s policy parameters, allowing us to obtain convergence to a recursive Stackelberg equilibrium in discrete/continuous state and discrete action space games. Our assumptions, proof techniques, and results imply convergence of policy gradient methods in a new class of MDPs, and open the door to proving the convergence of policy gradient methods in a new class of continuous/discrete state and discrete action space Markov games.
Even more importantly, we provide one of the first polynomial-time convergence guarantees for the well-studied class of reach-avoid problems [1] by modeling these problems as convex-concave zero-sum stochastic Stackelberg games. We are hopeful that our techniques might be picked up by roboticists who build robots to navigate real-world environments.

2) Beyond convex-concave domains, convergence to a Stackelberg equilibrium is NP-hard [3]. As global convergence guarantees to (recursive) Stackelberg equilibrium have not been obtained for more general zero-sum (stochastic) Stackelberg games under the assumption of a stochastic first-order oracle, we chose to restrict attention to convex-concave domains. It may also be possible to prove a local convergence result; however, such a result would overlook the fact that the popular class of reach-avoid problems [1] can be naturally formulated as convex-concave zero-sum stochastic Stackelberg games (Appendix C, Theorem C.1.), implying that we provide efficient solutions to these problems. A number of other problems of interest have also been shown to satisfy the convex-concavity assumptions (see for instance [4] or [5]) further motivating the class of convex-concave zero-sum stochastic Stackelberg games.

**References**

[1] Jaime F. Fisac, Mo Chen, Claire J. Tomlin, and S. Shankar Sastry. Reach-avoid problems with time-varying dynamics, targets and constraints. In Proceedings of the 18th International
Conference on Hybrid Systems: Computation and Control, HSCC ’15, page 11–20, New York, NY, USA, 2015. Association for Computing Machinery.

[2] Goktas, Denizalp, and Amy Greenwald. "Convex-concave min-max Stackelberg games." Advances in Neural Information Processing Systems 34 (2021): 2991-3003.

[3] Tsaknakis, Ioannis, Mingyi Hong, and Shuzhong Zhang. "Minimax problems with coupled linear constraints: computational complexity, duality and solution methods." arXiv preprint arXiv:2110.11210 (2021).

[4] Badithela, Apurva, et al. "Synthesizing reactive test environments for autonomous systems: testing reach-avoid specifications with multi-commodity flows." 2023 IEEE International Conference on Robotics and Automation (ICRA). IEEE, 2023.

[5] Goktas, Denizalp, and Amy Greenwald. "Gradient Descent Ascent in Min-Max Stackelberg Games." arXiv preprint arXiv:2208.09690 (2022).

---

### Decision · Program_Chairs · 2023-09-21

**Decision:**

Accept (poster)

**Comment:**

A polynomial-time algorithm is developed for zerosum stochastic Stackelberg games in the convex-concave case (as exemplified by the reach-avoid game). The key contribution is to identify a nontrivial class of games that possesses the convex-concave property, which allows the theoretical analysis of policy gradient algorithms to carry over (which on its own is incremental without the structural result). After a thorough discussion among reviewers, we have reached a consensus that the structural result is novel and passes the acceptance bar by a small margin.